# Assessing phototoxic drug properties of hydrochlorothiazide using human skin biopsies

Mathias Hohl [1] ✉, Felix Götzinger[1,2,3], Simone Jäger[1], Lea Wagmann[4], Mert Tokcan [1], Thomas Tschernig[5], Jörg Reichrath[6], Jan M. Federspiel [7], Peter Boor [8], Markus R. Meyer [4], Felix Mahfoud [1,2,3,9] & Michael Böhm [1,9]

The diuretic drug hydrochlorothiazide (HCT) is associated with an increased risk of non-melanoma skin cancer upon UV exposure. The underlying cellular and molecular mechanisms behind this association remain elusive. Herein, a human skin model to assess the photocarcinogenic effects of HCT is established. Skin biopsies collected from human body donors are treated with HCT and irradiated with 300 mJ/cm$^2$ low dose UVA or UVB or with 5 J/cm$^2$ high dose UVA. In HCT-treated biopsies but not in control, low dose UVA irradiation results in activation and nuclear translocation of the tumor-suppressor protein p53 accompanied by an upregulated gene expression of p53-negative regulator *MDM2*. High dose UVA additionally provokes DNA damage and initiation of pro-inflammatory gene expression. In contrast, UVB induces pronounced DNA damage, p53 protein activation, gene expression of *MDM2* and inflammatory marker genes in both HCT-treated biopsies and untreated control. In summary, in HCT-treated skin biopsies, activation of the p53-MDM2 axis, induction of DNA damage, and inflammatory response depends on UVA-dosage and may influence skin carcinogenesis over time. This human model eliminates the need for animal testing and mitigates species difference, offering a valuable tool for future drug development and safety testing.

Pharmacoepidemiologic studies associated the use of the diuretic hydrochlorothiazide (HCT) to an increased risk of non-melanoma skin cancer[1]. The association, although disputed, led to a reduction in HCT prescriptions in Europe without an adequate substitution by other equipotent antihypertensives[2], potentially worsening blood pressure (BP) management in thousands of patients[3]. Among others, HCT is believed to have phototoxic properties, potentially enhancing the damages caused by ultraviolet A or B (UVA, UVB) radiation[4]. A phototoxic reaction is initiated upon exposure to UV-light, if photosensitizer, like drugs or their metabolites, are accumulated in the skin. Upon UV-light of the appropriate wavelength the photosensitizer absorbs the light energy to form an excited triplet, which either reacts with oxygen to form reactive oxygen species, or it covalently binds to tissue molecules. Both mechanisms

causing cellular damages to membranes, lipids and DNA[5]. As a sulfonamide with halogenated side chains, HCT absorbs photons in the UVA (320–400 nm) and UVB range (280–320 nm) transferring the absorbed energy to the surrounding tissues including DNA[5–9]. The clinical and preclinical associations between HCT, phototoxicity, and carcinogenesis are based on small-scale preclinical studies, clinical case reports, and epidemiologic studies[4,5]. However, case-controlled trials and meta-analysis of randomized BP trials have not established a definitive relationship between HCT and phototoxicity or skin cancer[10,11]. Given the conflicting evidence from randomized and non-randomized clinical trials, the current methods for testing the phototoxic properties of (cardiovascular) drugs in preclinical and clinical trials may be insufficient. Animal models are traditionally used in drug development, but

[1]Department of Internal Medicine III – Cardiology, Angiology and Intensive Care Medicine, Saarland University Hospital, Saarland University, Homburg, Germany. [2]Department of Cardiology, University Heart Center Basel, University Hospital Basel, Basel, Switzerland. [3]Department of Biomedicine, University Hospital Basel and University of Basel, Basel, Switzerland. [4]Department of Clinical and Experimental Toxicology & Pharmacology, Saarland University, Homburg, Germany. [5]Institute of Anatomy and Cellbiology, Saarland University Hospital, Saarland University, Homburg, Germany. [6]Department of Adult and Pediatric Dermatology, Venerology and Allergology Saarland University Hospital, Saarland University, Homburg, Germany. [7]Institute of Legal Medicine, Saarland University, Faculty of Medicine, Homburg, Germany. [8]Institute of Pathology, University Clinic, Aachen, Germany. [9]These authors contributed equally: Felix Mahfoud, Michael Böhm. ✉e-mail: mathias.hohl@uks.eu

they are limited by species differences and the growing ethical pressure to reduce animal testing. Therefore, there is a critical need to establish a simple, reproducible ex vivo model using human skin biopsies from deceased body donors. To characterize this model, we investigated the phototoxicity of HCT comparing its effects under 300 mJ/cm² UVA or UVB and 5 J/cm² UVA radiation. UV radiation is absorbed by the skin, causing direct phototoxic DNA damage (UVB) or indirectly provoking oxidative cellular damage (UVA) both of which can contribute to carcinogenesis[12–14]. The tumor suppressor protein p53 and its negative regulator MDM2 are involved in the regulation of cell cycle arrest, DNA repair, cell growth and apoptosis[15–19]. Mutations in p53 are directly involved in the pathogenesis of UV-induced skin cancer and p53 induction is recognized as an early event in carcinogenesis[20]. UV-inducibility of histone H2AX phosphorylation (γH2A.X), a highly sensitive marker for DNA damage, regulation of tumor suppressor p53, and activation of pro-inflammatory pathways[21–23] was examined herein. Using human skin biopsies, we demonstrated that in the presence of HCT, low dose UVA leads to an activation of the p53/MDM2 axis, while high dose UVA additionally induces DNA damage and inflammatory responses, known factors that contribute to the cancerogenic potential attributed to photosensitizing drugs[5]. We conclude that HCT might acts as a photosensitizer, mediating molecular mechanisms known to be involved in cutaneous photocarcinogenic processes.

## Results

### Model functionality and responsiveness of skin biopsies to UVA and UVB

The experimental procedure of retrieval, irradiation, and treatment of human skin biopsies are depicted in Fig. 1. The establishment of a model for assessing phototoxic substances in human skin biopsies requires a well described positive control to verify and validate the vitality of the preparation and the functionality of the experiments. 8-Methoxypsoralen (8-MOP) is photoactivated by UVA (known as PUVA) to become a highly potent photosensitizing agent with known mutagenic and carcinogenic properties which is widely used as a photo-chemotherapy to treat severe skin disorders[23–25]. In 8-MOP-treated biopsies, irradiation with low dose UVA (300 mJ/cm²) resulted in pronounced DNA damage, elevated p53 protein levels, increased p53 phosphorylation and nuclear translocation, while p53-negative regulator MDM2 mRNA expression was unaffected. Low dose UVA also triggered activation of the central regulator protein p38 mitogen-activated protein kinase (MAPK), upregulation of pro-inflammatory Tumor Necrosis Factor alpha (TNFα), and repression of tumor-suppressors vitamin D receptor (VDR) and purinergic receptor P2X7 (P2RX7) transcription. UVB irradiation produced similar effects as observed upon UVA (Supplementary Fig. 1–3 **and**

Supplementary Table 1). Following irradiation with high dose UVA (5 J/cm²), elevated p53 protein levels, p53 phosphorylation, and nuclear translocation was accompanied by an upregulation of the DNA damage marker γH2A.X, while MDM2 mRNA was repressed. Additionally, high-dose UVA triggered activation of p38 MAPK, upregulation of TNFα, and the regulatory signaling molecule Connective Tissue Growth Factor (CTGF) mRNA expression, while transcription of tumor-suppressors VDR, and P2RX7 were downregulated. These effects were more pronounced after 24 h (Supplementary Fig. 4–6, Supplementary Table 2). Responsiveness of 8-MOP-treated skin biopsies to UVA and UVB demonstrates vitality of skin biopsies as well as assay functionality and reproducibility.

### Low dose UVA (300 mJ/cm²) irradiation of HCT-treated skin biopsies induces stabilization and phosphorylation of tumor suppressor p53 independent of DNA damage

In HCT-treated biopsies, 300 mJ/cm² UVA irradiation led to significant increases in p53 protein levels ($2.5 \pm 0.4$ vs. $1.2 \pm 0.1$ IOD/GAPDH, $P = 0.0170$ vs. Ctrl+UVA), and p53 phosphorylation ($2.4 \pm 0.3$ vs. $1.1 \pm 0.1$ IOD/GAPDH, $P = 0.0042$ vs. Ctrl+UVA) within 6 h. DNA damage, as indicated by γH2A.X formation, was not observed in neither HCT-treated nor control biopsies (Fig. 2A–D). In HCT-treated skin biopsies, UVA also upregulated MDM2 mRNA expression, while it had no effect in Ctrl ($2.1 \pm 0.2$ vs. $1.1 \pm 0.1$ relative gene expression/GAPDH, $P = 0.0062$ vs. Ctrl+UVA) (Fig. 2E). By contrast, 300 mJ/cm² UVB irradiation significantly increased protein level and phosphorylation-status of p53 in Ctrl and HCT compared to unirradiated biopsies. Stabilization and activation of p53 was accompanied by an increased formation of γH2A.X in Ctrl and HCT. There was no statistical difference between Ctrl+UVB compared to HCT + UVB (Fig. 2A–D). Gene expression of p53-regulator MDM2 was unaffected by UVB after 6 h in both groups (Fig. 2E).

At 24 h, UVA exposure resulted in elevated p53 protein levels ($2.5 \pm 0.1$ vs. $1.3 \pm 0.1$ IOD/GAPDH, $P = 0.0003$ vs. Ctrl+UVA) and increased p53 phosphorylation ($3.1 \pm 0.4$ vs. $1.2 \pm 0.1$ IOD/GAPDH, $P = 0.0020$ vs. Ctrl+UVA) in HCT-treated biopsies but not in Ctrl. In HCT + UVA, increase in p53 protein was even more pronounced than in HCT + UVB ($P = 0.0364$). Expression of γH2A.X was still unaffected by UVA in HCT and Ctrl (Fig. 2F–I). Activation of MDM2 mRNA expression was restricted to HCT + UVA, while it had no effect in Ctrl ($1.8 \pm 0.1$ vs. $1.2 \pm 0.1$ relative gene expression/GAPDH, $P = 0.0110$ vs. Ctrl+UVA) (Fig. 2J). Following UVB, p53, phospho-p53, γH2A.X, and MDM2 gene expression were increased in Ctrl and HCT compared to unirradiated biopsies (Fig. 2F–J). In the absence of UV light, HCT alone had no effect compared to untreated Ctrl throughout the experiments.

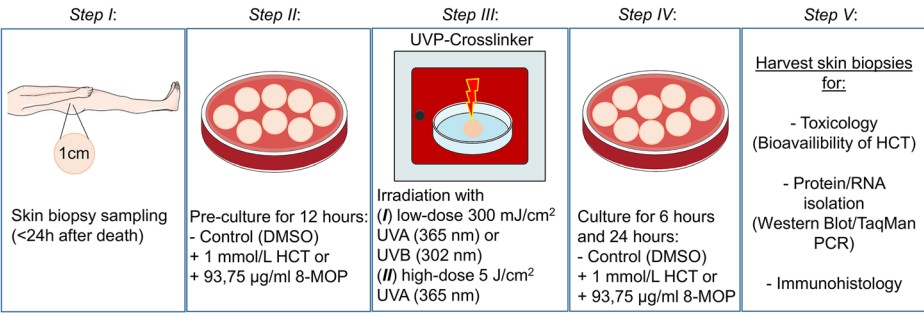

**Fig. 1 | Experimental procedure.** Skin biopsies were collected <24 h post-mortem from the hip of body donors. After sampling, biopsies were pre-incubated with either 1 mmol/L hydrochlorothiazide (HCT) dissolved in dimethyl sulfoxide (DMSO) or vehicle (DMSO, untreated control, Ctrl). 8-Methoxypsoralen (8-MOP, 93,75 μg/ml) was used as positive control. After 12 h, skin biopsies were placed into an UVP-crosslinker and irradiated from above without lid or cell culture medium with (**I**) either low dose 300 mJ/cm² of ultraviolet radiation type A (UVA; 365 nm) or B (UVB; 302 nm) or with (**II**) high dose 5 J/cm² UVA. Following irradiation, biopsies were incubated with either HCT, vehicle, or 8-MOP accordingly. Unirradiated biopsies served as group-specific control. Six and 24 h after irradiation, skin biopsies were harvested and processed for toxicology, isolation of total RNA and protein, as well as for immunohistology. Figure 1 was drawn using some schematic art images modified from Servier Medical Art, Copyright © 2006, Les Laboratories Servier (https://smart.servier.com).

**Fig. 2 | Low dose UVA irradiation caused increased p53 protein levels and enhanced phosphorylation and MDM2 mRNA expression in HCT-treated skin biopsies independent of γH2A.X. A** Representative Western blots demonstrating protein level of tumor suppressor protein p53, phosphorylation of histone H2A.X (γH2A.X, Serin139) and phosphorylation of p53 (Serin15) in untreated control biopsies (Ctrl), and in biopsies treated with HCT 6 h after irradiation with 300 mJ/cm² UVA or UVB. Unirradiated biopsies served as group-specific control (non). Protein expression of Glyceraldehyde 3-Phosphate Dehydrogenase (GAPDH) served as loading control. Quantification of (**B**) p53 protein (**C**) phospho-p53, and (**D**) γH2A.X. **E** Gene expression of p53-regulator MDM2 normalized against GAPDH 6 h after irradiation. **F** Representative Western blots of p53, γH2A.X and phospho-p53 (Serin15) in untreated control biopsies (Ctrl), and in biopsies treated with HCT 24 h after irradiation with 300 mJ/cm² UVA or UVB. Unirradiated biopsies served as group-specific control (non). Protein expression of GAPDH served as loading control. Quantification of (**G**) p53 protein (**H**) phospho-p53, and (**I**) γH2A.X. **J** Gene expression of MDM2 normalized against GAPDH 24 h after irradiation. For (**B, C, D, E, G, H, I, J**) $n = 6$ biopsies per group. Data are shown as mean ± SEM with individual points. For comparison of three groups, P-value was determined using Kruskal-Wallis with Dunn´s multiple comparisons test for Ctrl group in (**B, C, D, G, H, I**) and One-way ANOVA with Tukey multiple comparisons test for HCT group in (**B, C, D, G, H, I**). For (**E** and **J**) One-way ANOVA with Tukey multiple comparisons test was used. An unpaired student t-test (#) was used for comparison of 2 groups for (**B, C, E, G, H, J**). IOD: Integrated optical density. Numerical source data are provided within the Supplementary Data 1 file.

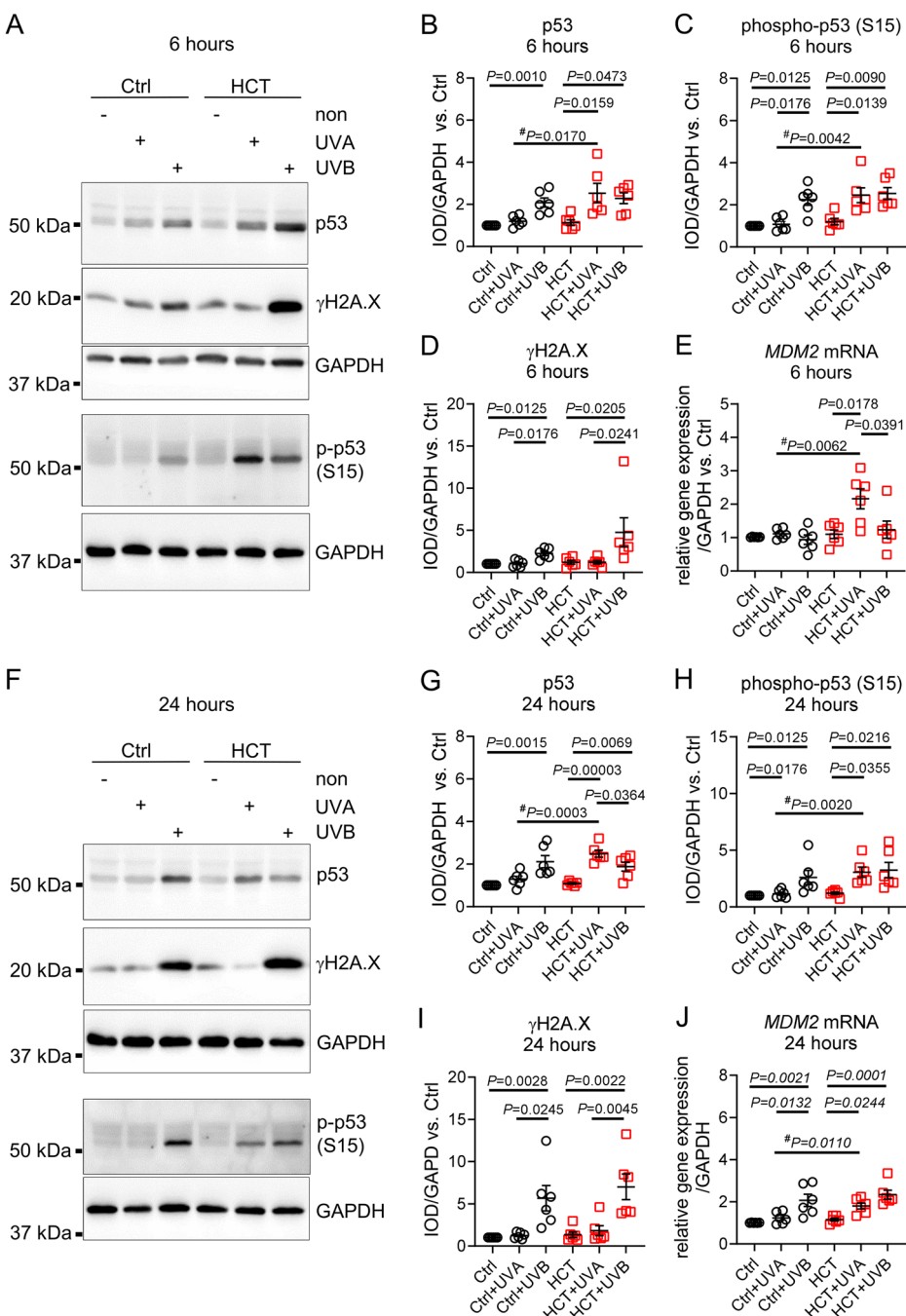

## Translocation of p53 in HCT-treated skin biopsies following low dose UVA irradiation

Human skin biopsies were subjected to immunofluorescence staining for γH2A.X or p53. To evaluate DNA damage and nuclear translocation of p53, the percentage of epidermal cells positive for γH2A.X or p53 was counted (Fig. 3A, B). 300 mJ/cm² UVA irradiation had no effect on the amount of γH2A.X-positive nuclei neither in HCT nor in Ctrl (Fig. 3C), but UVA triggered nuclear p53 translocation in HCT (after 6 hours: 29.3 ± 7.5 vs. 7.8 ± 3.1%, $P = 0.0297$ vs. Crtl+UVA; after 24 hours: 35.7 ± 4.0 vs. 6.2 ± 2.0%, $P = 0.0009$ vs. Crtl+UVA), while it had no effect in Ctrl (Fig. 3D). Following UVB irradiation, both the percentage of γH2A.X-positive nuclei and of p53-positive nuclei increased in both groups and at both time-points compared to unirradiated biopsies (Figs. 3C, D). Co-staining experiments demonstrated that γH2A.X and

p53 only co-localized within the nucleus following UVB irradiation but not UVA (Fig. 3E).

## Low dose UVA did not activate p38 MAPK nor induce inflammatory response in HCT-treated skin biopsies

Low dose UVA (300 mJ/cm²) had no effect on p38 MAPK phosphorylation or gene expression of pro-inflammatory markers such as Tumor Necrosis Factor alpha (TNFα), interleukin 6 (IL6) or signaling molecule Connective Tissue Growth Factor (CTGF) in either group after six hours (Fig. 4A–D). In contrast, UVB irradiation led to significant p38 MAPK phosphorylation in both groups, along with increased TNFα mRNA expression, while IL6 remained unchanged at this time-point. Expression of CTGF mRNA was increased upon UVB exposure in HCT-treated biopsies but not in Ctrl (Fig. 4A–D). At 24 h, UVA did not affect p38 MAPK phosphorylation or

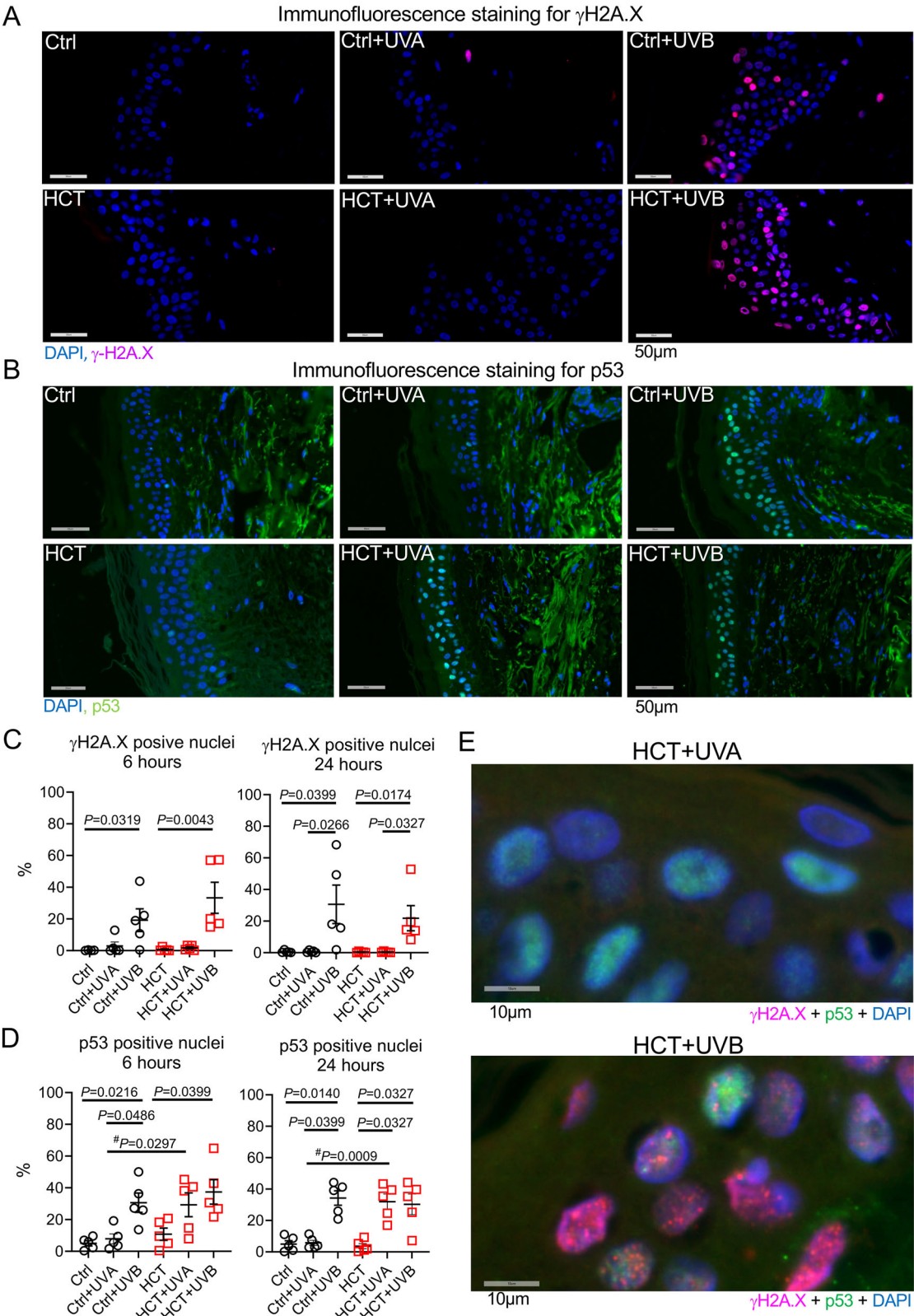

**Fig. 3 | Increased nuclear location of p53 in HCT-treated biopsies after low dose UVA irradiation in the absence of DNA damage.** Representative immunostaining for γH2A.X and p53 of human skin biopsies 6 h after UVA (**A**) or UVB (**B**) irradiation. **C** Quantification of γH2A.X positive stained nuclei 6 h and 24 h after UVA and UVB irradiation in the epidermis of Ctrl or HCT-treated skin biopsies. **D** Quantification of p53 positive stained nuclei 6 h and 24 h after UVA and UVB irradiation in the epidermis of Ctrl or HCT-treated skin biopsies. **E** Double-staining

for γH2A.X and p53. Data are shown as mean ± SEM with individual data points. For (**C**) n = 4 biopsies for Ctrl 6 h and n = 5 biopsies for all other groups. For (**D**) n = 5 biopsies per group. For comparison of three groups, P-value was determined using Kruskal-Wallis with Dunn´s multiple comparisons test for (**C** and **D**). An unpaired student t-test (#) was used for comparison of 2 groups for (**D**). IOD: Integrated optical density. Numerical source data are provided within the Supplementary Data 1 file.

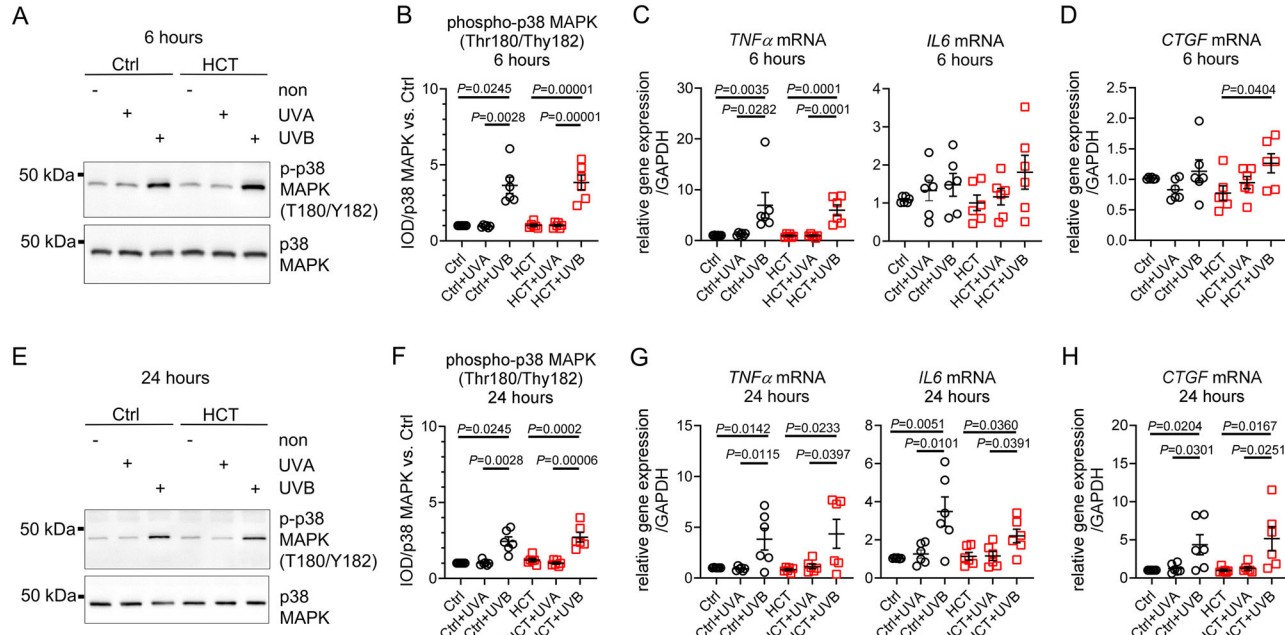

**Fig. 4 | Irradiation with 300 mJ/cm$^2$ UVB but not UVA activated p38 MAPK and pro-inflammatory response in HCT-treated skin biopsies. A** Representative Western blots demonstrating phosphorylation of p38 MAPK (T180/Y182) and total p38 MAPK protein in untreated control biopsies (Ctrl) and in biopsies treated with HCT 6 h after irradiation with 300 mJ/cm$^2$ UVA or UVB. Unirradiated biopsies served as group-specific control (non). **B** Quantification of phospho-p38 MAPK in relation to total p38 MAPK protein. **C** Gene expression of pro-inflammatory marker Tumor Necrosis Factor alpha (*TNFα*) and Interleukin 6 (*IL6*) and (**D**) Connective Tissue Growth Factor (*CTGF*) normalized against Glyceraldehyde 3-Phosphate Dehydrogenase (*GAPDH*) 6 h after irradiation. **E** Representative Western blots demonstrating phosphorylation of p38 MAPK (T180/Y182) and total p38 MAPK protein in Ctrl, or HCT 24 hours after irradiation with 300 mJ/cm$^2$ UVA or UVB.

Unirradiated biopsies served as group-specific control. **F** Quantification of phospho-p38 MAPK in relation to total p38 MAPK protein. **G** Gene expression of pro-inflammatory marker Tumor Necrosis Factor alpha (*TNFα*) and Interleukin 6 (*IL6*) and (**H**) Connective Tissue Growth Factor (*CTGF*) normalized against *GAPDH* 24 h after irradiation. For (**B, C, D, F, G, H**) $n = 6$ biopsies per group. Data are shown as mean ± SEM, with individual data points. For comparison of three groups $P$-value was determined using Kruskal-Wallis with Dunn´s multiple comparisons test for Ctrl group in (**B** and **F**) and an One-way ANOVA with Tukey multiple comparisons test for HCT group in (**B** and **F**). For (**C, D, G** and **H**) an One-way ANOVA with Tukey multiple comparisons test was used. IOD: Integrated optical density. Numerical source data are provided within the Supplementary Data 1 file.

mRNA expression of *TNFα*, *IL6* or *CTGF*, whereas UVB continued to upregulate p38 MAPK phosphorylation, and mRNA expression of *TNFα*, *IL6* and *CTGF* in both groups (Fig. 4E–H). Protein expression of anti-oxidative enzymes catalase, superoxide dismutase 1 (SOD 1) or SOD 2 was unchanged in all groups at all time-points (Supplementary Fig. 7).

### Irradiation of HCT-treated skin biopsies with high dose UVA (5 J/cm$^2$) results in stabilization of tumor suppressor p53, pronounced DNA damage and activation of inflammatory response

Six hours after irradiation with 5 J/cm$^2$ UVA, p53 protein stabilization and phosphorylation was induced, being more distinct in HCT-treated biopsies than in Ctrl. Following high dose UVA, expression of DNA damage marker protein γH2A.X was markedly increased in HCT-treated biopsies but not in Ctrl (1.25 ± 0.18 vs. 9.79 ± 2.4 IOD/GAPDH; $P = 0.1000$ vs. Ctrl+UVA) (Fig. 5A-D). Gene expression of p53-regulator *MDM2* was enhanced in Ctrl but not in HCT (Fig. 5E). Histological immunofluorescence staining for p53 and γH2A.X demonstrated an elevated number of p53-positive nuclei in Ctrl and HCT-treated biopsies after 5 J/cm$^2$ UVA irradiation, while γH2A.X-positive nuclei were increased only in HCT but not in Ctrl (0.90 ± 0.7 vs. 94.53 ± 2.5%; $P = 0.1000$ vs. Ctrl+UVA) (Fig. 5F–H, Supplementary Fig. 8). Following 5 J/cm$^2$ UVA, increased phosphorylation of p38 MAPK was accompanied by expression of *TNFα* and *CTGF* in HCT-treated biopsies but not in Ctrl (Fig. 5I–L).

Twenty-four hours after high-dose UVA irradiation, p53 protein, and phosphorylation of p53 were upregulated in Ctrl and, more pronounced, in HCT-treated skin biopsies. UVA-induced DNA damage, determined by elevated γH2A.X protein expression, was restricted to HCT (1.38 ± 0.25 vs. 6.91 ± 1.9 IOD/GAPDH, $P = 0.1000$ vs. Ctrl+UVA) (Fig. 6A–D). Following

UVA, expression of *MDM2* mRNA was activated in HCT and in Ctrl (Fig. 6E). Histological immunofluorescence staining for p53 and γH2A.X demonstrated an increased number of p53-positive nuclei in Ctrl and HCT-treated biopsies after UVA, while nuclei positive for γH2A.X were detected only in HCT + UVA (1.23 ± 0.6 vs. 88.13 ± 5.9%, $P = 0.1000$ vs. Ctrl+UVA) (Fig. 6F–H, Supplementary Fig. 9). In HCT but not in Ctrl, high dose UVA induced activation of p38 MAPK accompanied by an increased mRNA expression of pro-inflammatory *TNFα*, *IL6*, *IL1β*, and the signaling molecule *CTGF* (Fig. 6I–L). Irradiation with high-dose UVA had no effect on protein expression of anti-oxidative enzymes catalase, superoxide dismutase 1 (SOD 1), or SOD 2 in all groups at all time-points (Supplementary Fig. 10).

### Dose dependent effect of UVA on the expression of pro-apoptotic and carcinogenesis marker in HCT-treated skin biopsies

Low dose UVA (300mJ/cm$^2$) had no effect on the mRNA expression of the vitamin D receptor (*VDR*) or P2X7 receptor (*P2XR7*), neither in Ctrl nor in HCT (Table 1). However, in HCT-treated biopsies, irradiation with high dose UVA (5 J/cm$^2$) resulted in downregulation of *VDR* mRNA (1.09 ± 0.15 vs. 0.48 ± 0.03 relative gene expression; $P = 0.5264$ vs. Ctrl+UVA) and *P2XR7* mRNA (1.60 ± 0.21 vs. 0.38 ± 0.07 relative gene expression; $P = 0.0389$ vs. Ctrl+UVA) after 6 h. The observed UVA-induced effects were weakened after 24 h (Supplementary Table 3). Interestingly, pro-apoptotic *Bax* mRNA demonstrated a differential gene expression pattern depending on UVA-dosage. Six hours after low dose UVA-exposure, *Bax* mRNA was upregulated in HCT compared to Ctrl (1.10 ± 0.1 vs. 1.87 ± 0.3 relative gene expression/*GAPDH*, $P = 0.1508$ vs. Ctrl+UVA; $n = 5$ biopsies/

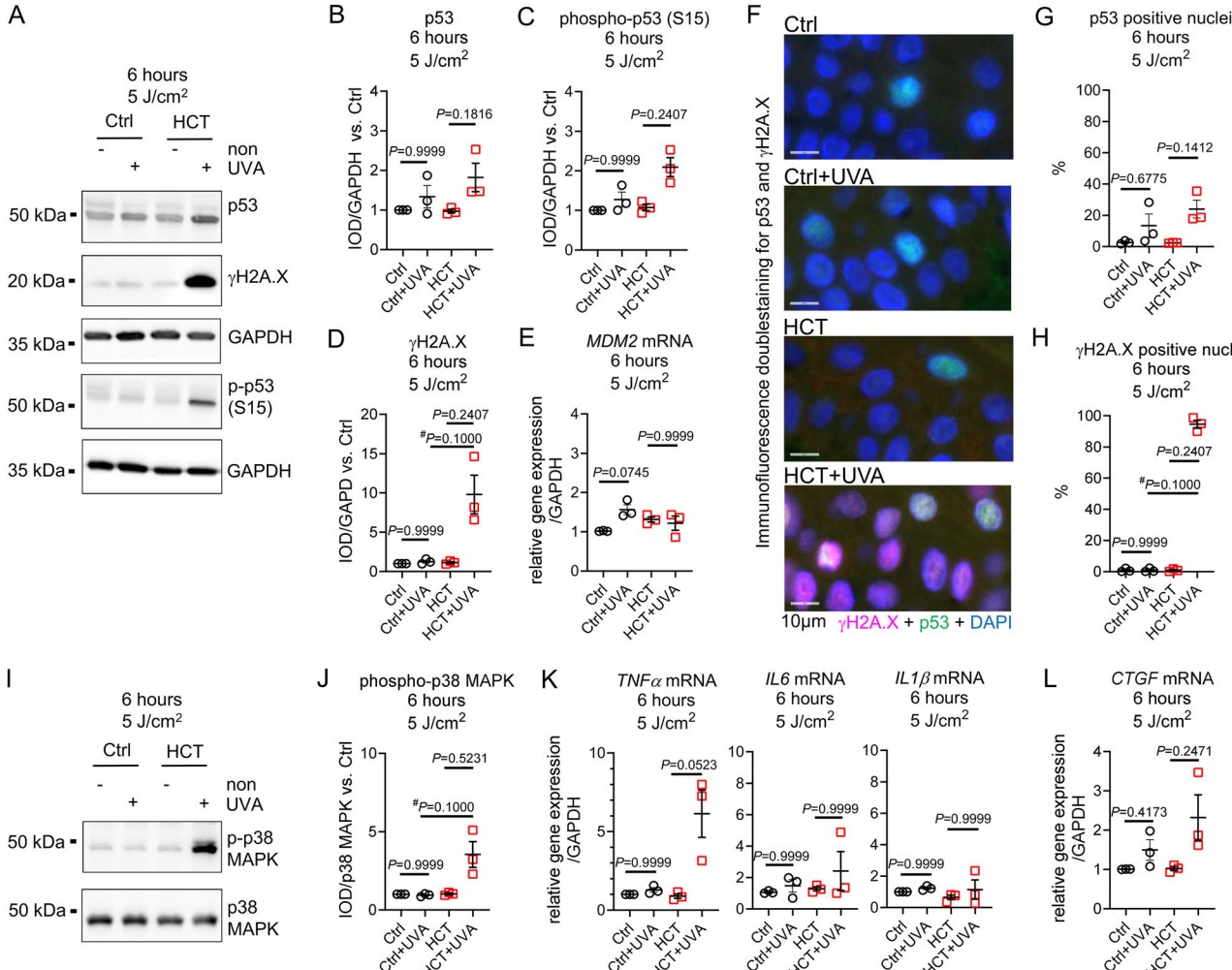

**Fig. 5 | UVA irradiation with 5 J/cm² induced p53 phosphorylation, nuclear translocation, γH2A.X formation, and inflammatory response in HCT-treated skin biopsies after six hours. A** Representative Western blots demonstrating protein level of tumor suppressor protein p53, phosphorylation of histone H2A.X (γH2A.X, Serin139) and phosphorylation of p53 (Serin15) in untreated control biopsies (Ctrl), and in biopsies treated with HCT six hours after irradiation with 5 J/cm2 UVA. Unirradiated biopsies served as group-specific control (non). Protein expression of Glyceraldehyde 3-Phosphate Dehydrogenase (GAPDH) served as loading control. Quantification of (**B**) p53 protein, (**C**) phospho-p53 (Ser15), and (**D**) DNA damage marker γH2A.X. **E** Gene expression of p53-regulator *MDM2* normalized against *GAPDH* 6 h after irradiation (**F**) Representative double-stained immunohistochemistry for γH2A.X and p53 of human skin biopsies 6 h after irradiation with 5 J/cm2 UVA. **G** Quantification of p53 positive stained nuclei and (**H**) γH2A.X positive stained nuclei 6 h after UVA irradiation in the epidermis of Ctrl or HCT-treated skin biopsies. **I** Representative Western blots demonstrating phosphorylation of p38 MAPK (T180/Y182) and total p38 MAPK protein in Ctrl, or HCT 6 hours after irradiation with 5 J/cm2 UVA. Unirradiated biopsies served as group-specific control (non). **J** Quantification of phospho-p38 MAPK in relation to total p38 MAPK protein. **K** Gene expression of pro-inflammatory marker Tumor Necrosis Factor alpha (*TNFα*), Interleukin 6 (*IL6*), Interleukin 1β (*IL1β*), and (**L**) Connective Tissue Growth Factor (*CTGF*) normalized against *GAPDH* 6 h after irradiation. For (**B, C, D, E, G, H, J, K, L**) *n* = 3 biopsies per group. Data are shown as mean ± SEM with individual points. For comparison of groups, *P*-value was determined using Kruskal-Wallis with Dunn´s multiple comparisons test for all groups in (**B, C, D, E, G, H, J, K, L**). A Mann-Whitney test (#) was used for comparison of 2 groups for (**D, H, J**). IOD: Integrated optical density. Numerical source data are provided within the Supplementary Data 1 file.

group), while *Bax* gene expression was significantly lowered in HCT after irradiation with 5 J/cm² compared to Ctrl (1.48 ± 0.14 vs. 0.73 ± 0.12 relative gene expression/*GAPDH*; *P* = 0.0191 vs. Ctrl+UVA; *n* = 3 biopsies/group). Following UVB irradiation, *VDR* and *P2XR7* gene expression was significantly repressed in Ctrl and HCT, with no additive effect of HCT. Anti-apoptotic *Bcl-2*, pro-apoptotic *Bak-1* and *Bax* mRNA were unaffected by UVB irradiation (Table 1).

## Discussion

This study successfully developed a model for phototoxicity testing and assessed the phototoxic properties of HCT using human skin biopsies irradiated with UVA or UVB. The main findings are the following: i) Human skin biopsies from body donors maintained responsiveness to UV irradiation and provided a reliable model for phototoxicity testing. ii) In

HCT-treated biopsies exposed to low dose UVA, stabilization and nuclear translocation of p53, as well as activated *MDM2* gene expression was independent of DNA damage or initiation of pro-inflammatory pathways. iii) In HCT, high dose UVA induced DNA damage, p53 activation, p38 MAPK phosphorylation, inflammatory gene expression and repressed mRNA transcription of vitamin D receptor (*VDR*) and purinergic receptor P2X7 (*P2XR7*). iv) Irradiation with 300 mJ/cm² UVB resulted in comparable effects as observed with 5 J/cm² UVA. v) HCT did not potentiate the photo-carcinogenic effects of UVB. Here, we report that in the presence of HCT, low dose UVA activates the p53/MDM2 axis, a critical mechanism involved in cell cycle regulation and apoptosis[15,16], while high dose UVA additionally induces DNA damage and inflammatory response. Thus, HCT may acts as a photosensitizer, mediating molecular mechanisms known to be involved in cutaneous photo-carcinogenic processes[26].

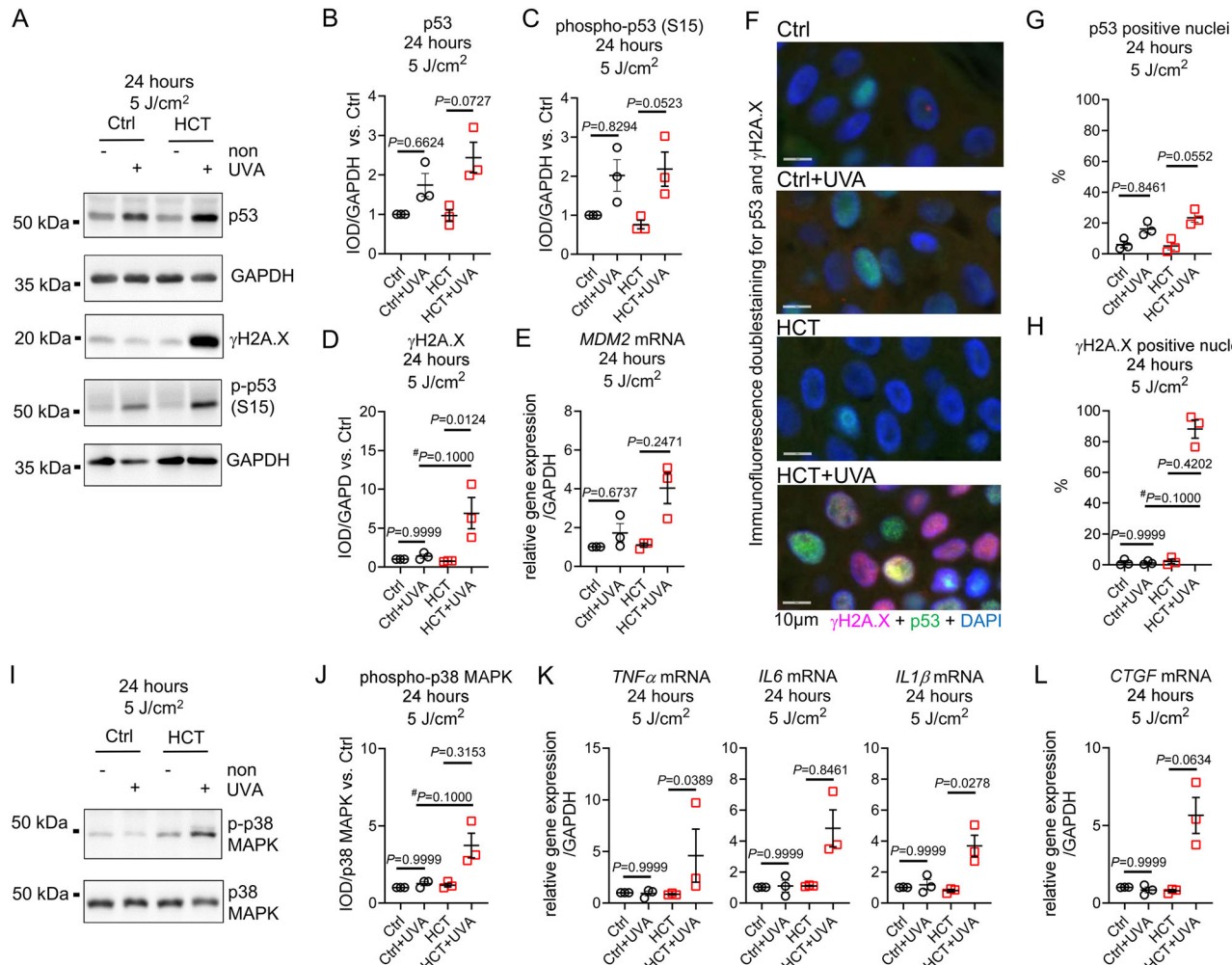

**Fig. 6 | UVA irradiation with 5 J/cm² induced p53 activation, pronounced γH2A.X formation and inflammatory response in HCT-treated skin biopsies after 24 h. A** Representative Western blots demonstrating protein level of tumor suppressor protein p53, phosphorylation of histone H2A.X (γH2A.X, Serin139) and phosphorylation of p53 (Serin15) in untreated control biopsies (Ctrl), and in biopsies treated with HCT 24 h after irradiation with 5 J/cm² UVA. Unirradiated biopsies served as group-specific control (non). Protein expression of Glyceraldehyde 3-Phosphate Dehydrogenase (GAPDH) served as loading control. Quantification of (**B**) p53 protein, (**C**) phospho-p53 (Ser15), and (**D**) DNA damage marker γH2A.X. **E** Gene expression of p53-regulator *MDM2* normalized against *GAPDH* 24 h after irradiation. **F** Representative double-stained immunohistochemistry for γH2A.X and p53 of human skin biopsies 24 h after 5 J/cm² UVA. **G** Quantification of p53 positive stained nuclei and (**H**) γH2A.X positive stained nuclei 24 h after UVA irradiation in the epidermis of Ctrl or HCT-treated skin

biopsies. **I** Representative Western blots demonstrating phosphorylation of p38 MAPK (T180/Y182) and total p38 MAPK protein in Ctrl, or HCT 24 h after irradiation with 5 J/cm2 UVA. Unirradiated biopsies served as group-specific control (non). **J** Quantification of phospho-p38 MAPK in relation to total p38 MAPK protein. **K** Gene expression of pro-inflammatory marker Tumor Necrosis Factor alpha (*TNFα*), Interleukin 6 (*IL6*), Interleukin 1β (*IL1β*), and (**L**) Connective Tissue Growth Factor (*CTGF*) normalized against *GAPDH* 24 h after irradiation. Data are shown as mean ± SEM with individual points. For (**B, C, D, E, G, H, J, K, L**) *n* = 3 biopsies per group. For comparison of groups *P*-value was determined using Kruskal-Wallis with Dunn's multiple comparisons test for (**B, C, D, E, G, H, J, K, L**). A Mann-Whitney test (#) was used for comparison of 2 groups for (**D, H, J**). IOD: Integrated optical density. Numerical source data are provided within the Supplementary Data 1 file.

UVA (320–400 nm) penetrates deep into the epidermis and dermis, causing genotoxic effects through indirect oxidative DNA damage[12,13]. By contrast, UVB (290–320 nm) is absorbed primarily by the skin epidermis, where its highly energetic photons are directly absorbed by DNA-bases, causing phototoxic damage. UV-induced DNA damage signals phosphorylation of histone H2AX (γH2A.X), a highly sensitive marker for DNA strand breaks, formation of DNA adducts, oxidative DNA damage and repair[21–23]. The tumor suppressor p53 plays a crucial role in regulating cell cycle arrest, DNA repair, cell growth and apoptosis[15,16]. A strict governance of p53 through interaction with its negative regulator MDM2 is critical for normal cell growth and development. In response to cellular stress signaling, like oxidative signaling[27] or DNA damage[17,27], p53 is rapidly phosphorylated which abolishes binding to MDM2 and leads to quickly elevated p53 levels due to reduced MDM2-mediated ubiquitination and degradation[17–19]. As an

early responder to DNA-lesions, p53 co-localizes with γH2A.X at DNA damage sites to orchestrate the DNA damage response cascade[20]. Cell culture experiments demonstrated that upon UV-irradiation, p53 is rapidly stabilized, followed by a later increase in MDM2 protein and mRNA levels, reversing p53 activity. This process is regulated in a temporal fashion, so that cell cycle progression can be halted while DNA repair continues prior to reversal of p53-mediated arrest by MDM2[28]. In line with this observation, in human skin biopsies treated with HCT, exposure to 5 J/cm² UVA or 300 mJ/cm² UVB led to DNA damage and an early increase in p53 protein levels, followed by a timely delayed activation of *MDM2* gene expression after 24 h. However, low dose UVA irradiation resulted in an early upregulation of p53 protein and *MDM2* gene expression while DNA damage was absent. This contrasts with the phototoxic 8-MOP control, which directly intercalate between nucleic acid base pairs[24], where low dose UVA-induced DNA

**Table 1 | Altered mRNA expression of marker genes involved in apoptosis and tumorigenesis following UV-irradiation**

| | Ctrl | Ctrl + UVA | Ctrl + UVB | HCT | HCT + UVA | HCT + UVB |
|---|---|---|---|---|---|---|
| Relative gene expression/GAPDH | 6 h | | | | | |
| Vitamin D-Receptor (VDR) | 1.01 ± 0.001 | 0.93 ± 0.05 | 0.60 ± 0.1 $P = 0.0006$ vs. Ctrl $P = 0.0051$ vs. Ctrl+UVA | 0.91 ± 0.09 | 0.91 ± 0.07 | 0.62 ± 0.1 $P = 0.0380$ vs. HCT $P = 0.0403$ vs. HCT + UVA |
| P2X7- Receptor (P2RX7) | 1.02 ± 0.005 | 0.96 ± 0.05 | 0.33 ± 0.1 $P = 0.0034$ vs. Ctrl $P = 0.0281$ vs. Ctrl+UVA | 0.86 ± 0.1 | 1.05 ± 0.1 | 0.39 ± 0.1 $P = 0.0205$ vs. HCT $P = 0.0016$ vs. HCT + UVA |
| Bcl-2 | 1.01 ± 0.01 | 1.11 ± 0.2 | 0.98 ± 0.06 | 1.10 ± 0.2 | 1.17 ± 0.3 | 1.14 ± 0.2 |
| Bak1 | 1.02 ± 0.01 | 1.25 ± 0.2 | 0.75 ± 0.1 | 1.55 ± 0.4 | 1.74 ± 0.4 | 1.38 ± 0.4 |
| Bax | 1.01 ± 0.006 | 1.10 ± 0.1 | 0.97 ± 0.1 | 1.17 ± 0.2 | 1.87 ± 0.3 | 1.35 ± 0.2 |
| Relative gene expression/GAPDH | 24 h | | | | | |
| Vitamin D-Receptor (VDR) | 1.00 ± 0.002 | 1.05 ± 0.09 | 0.75 ± 0.06 $P = 0.0282$ vs. Ctrl $P = 0.0117$ vs. Ctrl+UVA | 0.95 ± 0.08 | 0.99 ± 0.06 | 0.76 ± 0.09 |
| P2X7- Receptor (P2RX7) | 1.00 ± 0.001 | 1.09 ± 0.01 | 0.51 ± 0.1 $P = 0.0043$ vs. Ctrl+UVA | 1.17 ± 0.2 | 1.09 ± 0.04 | 0.57 ± 0.1 $P = 0.0040$ vs. HCT $P = 0.0109$ vs. HCT + UVA |
| Bcl-2 | 1.01 ± 0.002 | 0.86 ± 01 | 0.87 ± 0.2 | 0.94 ± 0.04 | 1.11 ± 0.1 | 0.87 ± 0.1 |
| Bak1 | 1.00 ± 0.002 | 1.20 ± 0.2 | 1.06 ± 0.2 | 1.01 ± 0.1 | 1.52 ± 0.4 | 1.08 ± 0.2 |
| Bax | 1.00 ± 0.002 | 1.02 ± 0.1 | 1.26 ± 0.2 | 1.10 ± 0.1 | 1.32 ± 0.1 | 1.49 ± 0.2 |

Data are shown as mean ± SEM. For vitamin D-Receptor (VDR) and P2X7-Receptor (P2RX7) $n = 6$ biopsies per group, for Bcl-2, Bak1, and Bax $n = 5$ biopsies per group. For comparison of three groups $P$-value was determined using Kruskal-Wallis with Dunn´s multiple comparisons test for Bcl-2, Bak1 and Bax in all groups and in Ctrl group to determine P2RX7 at 6 h and 24 h. One-way ANOVA with Tukey multiple comparisons test was used for VDR of all groups and for the HCT group determine P2RX7 at 6 and 24 h. Numerical source data are provided within the Supplementary Data 1 file. Gene names are written in italic.

damage was associated with increased p53 stabilization and repressed MDM2 gene expression. It is conceivable that in the long-run recurring UV-induced disruption of p53-MDM2 complex may have carcinogenic effects[29]. Imbalance of the p53-MDM2 axis has critical consequences, due to either chronically active p53 or persistent repression via MDM2. In mice, a hepatocyte-specific MDM2 deletion resulted in constitutively active p53. These mice developed multiple liver tumors with upregulated p53-target genes, increased hepatocyte apoptosis, senescence, and provoked liver inflammation[30,31]. Deregulation of MDM2 expression, mediates constitutive inhibition of p53 and uncoupling from p53-regulated growth control and thus might contribute to carcinogenesis independently of p53[32,33].

The carcinogenic action of UVA is generally attributed to oxidative DNA damage caused by endogenous cellular photosensitizers and direct photochemical production of cyclobutane pyrimidine dimers (CPD)[14]. Halogenated drugs like HCT may undergo photo-dehalogination, generating free radicals, further contributing to phototoxicity[9]. We observed a UVA-dose dependent phototoxicity fortified by HCT, with an early activation of the p53-MDM2 axis at low dose followed by pronounced DNA damage and inflammation at higher doses. In DNA-repair deficient xpa-knockout mice and wild type mice receiving HCT, irradiation with 30 J/cm2 UVA (365 nm) resulted in a significant increase in cyclobutane pyrimidine dimers (CPDs) formation in the epidermis, indicating genotoxic damage[8]. Also, in long-term cell culture experiments (9 weeks), exposure of HCT-treated human keratinocytes with 10 J/cm² UVA twice a week, was associated with increased genotoxic DNA damage, apoptosis resistance, chronic inflammation, and defective DNA-repair ability[34]. In vitro studies in melanocytes and fibroblasts demonstrated increased formation of reactive oxygen species and decreased cell viability in HCT-treated cells irradiated with UVA (5 J/cm²), confirming the phototoxic properties of HCT[35].

UVB irradiation is a potent inductor of photo-induced carcinogenesis independent of endogenous photosensitizers[14]. Indeed, UVB induced DNA damage, activation of tumor suppressor p53 and pro-inflammatory pathways were not aggravated by the presence of HCT. In DNA-repair deficient xpa-knockout mice, chronic exposure to broad-band UVB (275-390 nm, peak emission at 313 nm) alone increased the incidence of skin tumor formation. UVB-induced carcinogenesis was not fortified in the presence of HCT[36]. Lerche et al. (2022)[37] demonstrated that chronic exposure of hairless mice treated with different doses of HCT to broad-band UV radiation (UVR with 5.9% of the emitted wavelength was within the UVB range, wavelength maximum 365 nm) showed no differences in the time to the occurrence of the first, second or third skin cancer, as compared to hairless mice, that were only treated with UV radiation[37].

The p38 mitogen-activated protein kinase (MAPK) is activated by phosphorylation in response to DNA damage[38]. Activation of p38 MAPK is involved in the pro-inflammatory response of the skin following exposure to UV light[39]. Within this model, activation of p38 MAPK was accompanied with increased expression of TNFα and IL6 in a time-dependent manner. This effect could be observed following high dose UVA in the present of HCT and following UVB in both, control and HCT. It was reported that a single dose of UVB (360 mJ/cm²) induced epidermal p38 MAPK signaling and induced pro-inflammatory response in the skin of hairless mice[40]. Interestingly, in our model early activation of p38 MAPK was followed by a late upregulation of CTGF mRNA expression. CTGF protein is a negative regulator of p38 MAPK and CTGF-mediated dephosphorylation of activated p38 MAPK is involved in anti-apoptotic processes and protection against apoptosis[41,42].

Besides the ability of high dose UVA to induce DNA damage and initiate pro-inflammatory processes in HCT-treated biopsies, it was also associated with downregulation of the VDR and P2XR7 mRNA, an effect also observed after UVB irradiation in Ctrl and HCT. The vitamin D receptor acts as a tumor suppressor and its downregulation increases susceptibility to tumor formation in the skin[43–45]. The purinergic receptor P2X7 acts as an apoptosis-inducing receptor and is involved in the regulation of UV-induced apoptosis after unrepairable DNA damage or malfunction of the DNA-repair mechanisms[46]. Reduction of P2X7 receptor after UVB irradiation increases the probability of survival of malignant cells thereby possibly increasing tumor aggressiveness[47,48].

Pharmacoepidemiologic studies have suggested an association of long-term HCT use and an increased risk of non-melanoma skin cancers[1,4,49,50]. However, as these studies are observational and non-randomized, they are

prone to biases. A small randomized controlled trial and a large meta-analysis of randomized controlled trials did not show associations of HCT use and phototoxicity or skin cancer risk[10,11]. It is important to note that susceptibility to skin cancer can be influenced by skin type, (sun)light exposure, genetics, age, and other variables[51]. Drug-induced photo-sensitivity and phototoxicity is estimated to occur in 8% of all adverse drug reactions[4]. Since, sun tanning behavior and polypharmacy are increasingly practiced, these numbers might increase in the future. Patient management however can easily be impacted by advising sun protective behavior, like wearing of protective clothes, applying sunscreen and avoiding exposure when irradiation is the highest[51]. From a clinical perspective, the biggest impact of possible phototoxicity might be non-adherence to drugs because of fear of non-melanoma skin cancers[2]. As stated above, hypertension remains the most common cause of premature death and morbidity worldwide and blood pressure control reduces the incidence of stroke, heart failure, myocardial infarction and chronic kidney disease[52]. Not treating hypertension in fear of semi-malignant or non-malignant skin tumors might cause more harm than good to individuals. Moreover, phototoxicity might be easily avoidable by using protective clothes and sunscreen[52]. More evidence is needed to better understand the benefit from risk assessment between HCT, hypertension and non-melanoma skin cancers.

This study provides new mechanistic insights into the possible link between HCT and non-melanoma skin cancer, adding to the conflicting evidence on this topic. While our findings suggest a potential connection between HCT intake and skin carcinogenesis, these results are hypothesis-generating and should be confirmed through larger-scale and long-term exposure studies. Additionally, in this study we only used narrowband UVA and UVB. While a combination of both wavelengths would better simulate solar radiation, our data clearly show that the highly energetic UVB would overshadow the differential effects of UVA if the two were combined. Lower doses of radiation were used, as recommended in the German S1 Guidelines on UV phototherapy and photochemotherapy for PUVA therapy and as we observed reactions to lower doses of UVA in previous clinical trials[11,25].

The use of human skin biopsies bridges the gap between monolayer cell culture experiments and animal testing, as biopsies retain the epidermal and dermal structures, comprising various specialized cell types with preserved cell-cell contacts and structure. Administration of multiple drugs and repeated exposure to UV-radiation would more closely reflect in vivo conditions and are conceivable using human skin biopsies, if viability, structural integrity, and functional metabolism are maintained[53,54]. Cada-veric skin or skin biopsies in general, are detached from the central nervous system, from blood and lymphatic vessels, with vital effects on temperature regulation, blood flow, supply with nutrients and oxygen, the disposal of metabolic waste products, influx of inflammatory cells, wound healing processes or angiogenesis/lymphangiogenesis, the latter being involved in tumorigenesis[55,56]. Also, the study of UV-induced skin tumor formation in this model is limited by metabolic changes and the gradual loss of tissue integrity over time under cell culture conditions. Though, it has been reported that human explant skin is structurally viable and metabolically active for up to 9 days in culture[54,55].

Summarized, in human skin biopsies treated with HCT, exposure to a low dose of UVA activated the p53-MDM2-axis, a key regulatory pathway involved in cell cycle control and cell growth, independent of DNA damage. Exposure to a higher dosage of UVA, induced DNA-injury and inflam-matory response, adding to the uncoupling of p53 from growth control. Thus, the phototoxic properties of HCT might, over time, contribute to skin carcinogenesis. Furthermore, human skin biopsies from body donors pro-vide a viable model for investigating acute molecular mechanisms involved in the phototoxic reactions of cardiovascular drugs. This model offers sev-eral advantages, including the reduction of animal testing, the expansion of cell culture-based experiments, and minimizing the confounding effects of species differences, thus helping to bridge the translational gap between rodent studies and human applications. The human skin biopsy model might also be used in combination with already established in vitro assays, like the highly sensitive 3T3 NRU-phototoxicity test, the Reactive Oxygen Species (ROS)-assay for chemical photoreactivity, or the Reconstructed Human Epidermis Phototoxicity test[57,58].

## Methods

### Human skin model and ex vivo culture conditions

Body donors consented to body donation for scientific studies during their lifetime at the Anatomical Institute of Saarland University, Homburg/Saar, Germany. The ethical committee of the Medical Association of Saarland approved the study (number 162/20). All ethical regulations relevant to human body donors were followed. Skin biopsies (Six female, three male, median age: $84.56 \pm 3.2$ years, Caucasians with a skin type II-III on the Fitzpatrick scale) were collected <24 h after death from the upper hip of the body donor from skin areas that were macroscopic without pathological findings and without signs of putrefaction. Skin from the upper hip was chosen to reduce potential confounders based on UV-exposure during life-time. The hip was sterilized using antiseptic octeniderm® (Schülke & Mayr GmbH. Norderstedt, Germany) and biopsies ($n = 31$ per donor) were retrieved using a sterile 10.0 mm biopsy punch (SMI AG, St. Vith, Belgium, #ZBP10), forceps and scalpel (Präzisa plus, Dahlhausen & Co. GmbH, Köln, Germany). Supplementary Fig. 11 depicts a representative hematoxylin and eosin (H.E.) staining of a skin biopsy visualizing the squamous epithelium (comprising the stratum corneum, stratum lucidum, stratum granulosum, stratum spinosum, and stratrum basale), the epidermis and the dermis. One biopsy per body donor was shock frozen immediately in liquid nitrogen for toxicological analysis (blank probe). After sampling, biopsies were imme-diately transferred into a 50 ml collecting tube containing cell culture medium (Dulbecco's modified eagle medium (DMEM: gibco, life technol-ogies limited, United Kingdom), 10% fetal calf serum (FSC), and 1% streptomycin and penicillin). Under sterile conditions, biopsies were cut in half, transferred to 60 mm culture plates (TC dish, Sarstedt AG, Nümbrecht, Germany, #83.3901) containing 3 ml cell culture medium ensuring the air-liquid interface and cultured at 37 °C and 5% $CO_2$ (Heraeus Kendro, HERAcell Inkubator). Skin biopsies were incubated in the absence of light for 12 h with either 1 mmol/L HCT (Sigma-Aldrich, Steinheim, Deutsch-land; #H2915-5G) dissolved in DMSO, or 8-Methoxypsoralen (8-MOP; 93,75 µg/ml; positive control; Meladinine 0,75%, Medipha Sante, Courta-boeuf Cedex, France). 8-Methoxypsoralen (8-MOP) is considered a strong phototoxic and pro-carcinogenic chemical and is often used as a positive control for phototoxic assays[23,24]. Therefore, all experiments were conducted in parallel using skin biopsies treated with 8-MOP as a positive control. Untreated biopsies served as negative control (Ctrl; contained only DMSO) (Fig. 1).

### Irradiation of human skin biopsies

After 12 h, skin biopsies from six body donors (four female, two male) were transferred to a fresh and sterile 60 mm TC dish, placed into an UVP-crosslinker (Analytik Jena, CL-1000M, 8 W, Upland, USA) and irradiated from above without lid or cell culture medium with either low-dose 300 mJ/$cm^2$ of pure ultraviolet radiation type A (UVA; 365 nm) or B (UVB; 302 nm) to avoid UV-range interference[11,25]. In a second experimental approach, skin biopsies from three body donors (2 female, one male) were irradiated with high-dose 5 J/$cm^2$ UVA. Following irradiation, biopsies were transferred to sterile 60 mm dishes containing fresh cell culture medium supplemented accordingly with either HCT, 8-MOP or DMSO only (Ctrl) as described above. Unirradiated biopsies served as group-specific control. Six and 24 h after irradiation, skin biopsies were harvested and either shock frozen in liquid nitrogen for toxicologic analysis, fixed in buffered 4% formaldehyde for immunohistology or transferred to Tri Reagent® (Sigma-Aldrich, Steinheim, Germany) for isolation of total RNA and proteins.

### Quantification of HCT in human skin biopsies using ultra-high-performance liquid chromatography and high-resolution tandem mass spectrometry (LC-HRMS/MS)

Biopsies, comprising blank probes, treated and untreated probes during cell culture experiments, were washed three times in 0.9% sodium chloride-

solution (Fresenius Kabi AG, Bad Homburg, Germany), wiped dry (Kimtech science TM, precision wipes, Kimberly-Clark®, Ede, The Netherlands) and transferred into reaction tubes containing homogenization buffer (5 mmol/L EDTA; 25 mmol/L NaF; 300 mmol/L sucrose; 30 mmol/L KH2PO4, pH = 7.0) and complete protease inhibitors (#11873580001; Roche, Penzberg, Germany). Biopsies were homogenized using a Branson Sonifier 250 (Heinemann Ulraschall- und Labortechnik, Schwäbisch-Gmünd, Germany), centrifuged and the supernatant stored at $-20\,°C$ for ultra-high-performance liquid chromatography-high resolution mass spectrometry. After homogenization, 50 µL of the skin homogenates and 10 µL of the internal standard solution containing 13C6-HCT (10 mg/L in methanol) were added to a 1.5 mL reaction tube. The mixture was extracted with 1 mL of ethyl acetate by vortexing for 10 min. After centrifugation (5 min, 18,000 x g at 23 °C), 1 mL of the upper layer was transferred to a glass vial and evaporated to dryness under a gentle stream of nitrogen at 40 °C. The residue was reconstituted with 50 µL of eluents A and B (1:1, v/v, see below) and a 5 µL aliquot was analyzed by LC-HRMS/MS. A Dionex Ultimate UHPLC System (Thermo Fisher, TF, Dreieich, Germany) and a TF Accucore PhenylHexyl column (100 mm × 2.1 mm, 2.6 µm particle size) maintained at 40 °C were used for the chromatographic separation. The mobile phases consisted of eluent A (2 mM aqueous ammonium formate plus formic acid, 0.1%, v/v, pH 3) and eluent B (acetonitrile plus formic acid 0.1%, v/v). The flow rate was set to 0.6 ml/min. Starting mobile phase conditions were 98% A and 2% B; mobile phase B was increased to 98% from 0.1 to 5.0 min and held for 1.0 min. Afterward, mobile phase B was decreased to 2% from 6.0 to 7.0 min, and starting conditions were held for 0.5 min. Detection of analytes was achieved via high-resolution mass spectrometry on a TF Q Exactive plus system using negative electrospray ionization. The settings of the instrument were as follows: sheath gas, 50 arbitrary units (AU); auxiliary gas, 10 AU; spray voltage, $-4.00$ kV; heater temperature, 320 °C; ion transfer capillary temperature, 320 °C; and S-lens RF level, 60.0. MS for identification of the analytes was performed using negative full scan (FS) data and a product reaction monitoring (PRM) mode. The settings for FS data acquisition were as follows: resolution, 17,500; microscans, 1; automatic gain control (AGC) target, 3e6; maximum injection time (IT), 200 ms; scan range, m/z 150–2000. The settings for PRM were as follows: resolution, 17,500; microscans, 1; AGC target, 2e5; maximum IT, 100 ms; isolation window, 2.0 m/z, spectrum data type, profile; an inclusion list for the m/z of interest (295.9569 for HCT, HCD 35, and 301.9773 for 13C6-HCT, HCD 35;). 8-MOP was not detectable by LC-HRMS/MS due to poor ionization. The quantification of HCT was individually performed via the internal standard in each sample. All analytical runs consisted of blank and zero samples and the study samples.

HCT concentrations were analyzed at baseline prior to incubation, as well as 18 h and 32 h after incubation of biopsies with 1 mmol/L HCT. Levels increased from 0.06 ± 0.03 mg/L at baseline, to 33.3 ± 4.2 mg/L after 18 h, and 33.4 ± 4.4 mg/L after 36 h of incubation, confirming HCT bioavailability (Supplementary Fig. 12).

### Histology and immuno-histochemical staining
For an overview-staining biopsies were fixed in buffered 4% formaldehyde for 48 h and imbedded in paraffin for histological evaluation. Tissue sections of 3 µm were fixed at +56 °C overnight, deparaffinized, rehydrated and stained with hematoxylin and eosin (HE, Morphisto, Frankfurt am Main, Germany). An Aperio ImageScope x64 whole slide scanner (Leica biosystems, Aperio ImageScope Wetzlar, Germany) was used for image acquisition, analysis, and visualization.

Immunofluorescence-double staining of p53 and γH2A.X human skin biopsies: Following de-paraffinization and hydration as described above, 3 µm sections were incubated for 1 h in 0.05% citraconic anhydrid (Sigma-Aldrich Chemie GmbH, Munich, Germany; pH 6.8) at +95 °C in a water bath and washed afterwards twice in 1xPBS (phosphate-buffered saline: 137 mmol/L NaCl; 2.7 mmol/L KCl; 4.3 mmol/L Na2HPO4; 1.47 mmol/L KH2PO4, pH 7.4). Slices were incubated overnight with primary antibody (Anti-p53 (DO-7) mouse mAb, (#48818S; Cell Signaling Technologies),

diluted (1:100) in 1xPBS in a moisture-chamber at +4 °C. After washing twice with 1xPBS containing 0.1% Tween20 and once with 1xPBS, slices were incubated with the secondary antibody anti-mouse-IgG-FITC (#715-095-150, Dianova, Hamburg, Germany), diluted 1:50 in 1xPBS for 2 h at +37 °C. In a second staining step, slices were incubated overnight with γH2A.X primary antibody (Anti-γH2A.X (phospho S139) [EP854(2)Y], rabbit monoclonal Ab, Abcam, #ab81299), and diluted (1:100) in 1xPBS in a moisture-chamber at +4 °C. After washing twice with 1xPBS containing 0.1% Tween20 and once with 1xPBS, slices were incubated with the secondary antibody anti-rabbit-IgG-TRITC (#711-026-152; Dianova, Hamburg, Germany), diluted 1:50 in 1xPBS for 2 h at +37 °C. Slices were mounted with DAPI mounting medium (#H-1200, Vector Laboratories Inc, Burlingame, USA). For image acquisition, analysis, and visualization tool Aperio ImageScope x64 (Leica biosystems, Wetzlar, Germany) was used. For quantification of p53 and γH2A.X positive nuclei, up to 1000 DAPI-stained nuclei were counted and allocated with the number of p53 or γH2A.X positive nuclei.

### Western Blot and TaqMan PCR from human skin biopsies
Western Blot and TaqMan PCRs were performed in two biopsy samples per body donor. The mean value of these two replicates represent the value for each body donor. For isolation of total RNA and protein from the same biopsy, TRI Reagent® (Sigma-Aldrich, #93289-100 ml) was used following the manufacturer's protocol.

Western Blot: 50 µg of protein were separated on 10% SDS-PAGE and electrophoretically transferred to nitrocellulose membranes (0.2 µm pore size, #1620112, Bio-Rad Laboratories, Inc., Germany). Membranes were blocked in 1xPBS-T (Phosphate-Buffered Saline: 137 mmol/L NaCl; 2.7 mmol/L KCl; 4.3 mmol/L Na2HPO4; 1.47 mmol/L KH2PO4, pH 7.4 containing 0.1% Tween) containing 5% non-fat dry milk for at least 120 min at room temperature and exposed to the following primary antibodies overnight: Anti-γH2A.X (phospho S139), rabbit monoclonal Ab, Abcam, #ab81299, diluted 1:1000, anti-p53 rabbit Ab (#9282S; Cell Signaling Technologies, diluted 1:1000, anti-phospho-p53 (S15) rabbit Ab (#9284S; Cell Signaling Technologies, diluted 1:1000, anti-p38 MAPK rabbit Ab (#9212S; Cell Signaling Technologies, diluted 1:1000, anti-phospho-p38 MAPK (T180/Y182) (12F8) rabbit Ab (#4631S; Cell Signaling Technologies, diluted 1:1000, anti-Catalase rabbit mAb (#14097S, Cell Signaling Technologies, diluted (1:1000), anti-superoxide dismutase 1 (SOD-1) rabbit polyclonal IgG (sc-11407, Santa Cruz Biotechnology, diluted 1:5000), anti-superoxide dismutase 2 (SOD-2) rabbit polyclonal IgG (sc-30080, Santa Cruz Biotechnology, diluted 1:5000), anti-glyceraldehyde-3-phosphate dehydrogenase (GAPDH; MAB374, Millipore, Darmstadt, Germany, diluted (1:10000)). Respective secondary antibodies (purchased from Bio-Rad Laboratories, Inc., USA: anti-mouse: #170-6516, anti-rabbit: #172-1019; diluted 1:10000 in 1xPBS-T containing 0.5% non-fat dry milk) were incubated for 60 min at room temperature. Proteins were visualized by enhanced chemi-luminescence according to the manufacturer´s guidelines (#RPN2106, Amersham Pharmacia Biotech, Amersham, UK) and analyzed using the Vilber Loumat Fusion Solo S documentation system (Evolution-Capt Edge) (Avantor™, VWR™, Germany). Data are presented as integral optical density (IOD) normalized to GAPDH.

Polymerase Chain Reaction (PCR): Gene expression analysis was performed by TaqMan PCR. Following isolation of RNA according to the TRI Reagent®-protocol, genomic DNA impurities were removed by DNase treatment (Peqlab, Erlangen, Germany), and cDNA was synthesized by reverse transcription using the HighCap cDNA RT Kit (#4368814; Applied Biosystems, Waltham, USA) according to the manufacturer´s protocol. TaqMan PCR was conducted in a StepOne plus thermocycler (Applied Biosystems, Waltham, USA) using TaqMan GenEx Mastermix (#4369016, Applied Biosystems, Waltham, USA). Signals were normalized to GAPDH controls. No template controls were used to monitor for contaminating amplifications. The ΔCt was used for statistical analysis and $2^{-\Delta\Delta Ct}$ for data presentation. Human probes used to amplify the transcripts were as follows (purchased from Thermo Fisher Scientific): Interleukin 6 (*IL6*,

Hs00174131_m1), Tumor necrosis factor alpha (*TNFα*; Hs00174128_m1), Interleukin 1β (*IL1β*; Hs1555410_m1), Vitamin D receptor (*VDR*, Hs01045843_m1), P2X7 receptor (*P2RX7*; Hs00175721_m1), connective tissue growth factor (*CTGF*; Hs01026927_g1), *Bax* (Hs00180269_m1), *Bak-1* (Hs00832876_g1), *Bcl2* (Hs04986394_s1), *MDM2* (Hs01066930_m1), Glyceraldehyde 3-phosphate dehydrogenase (*GAPDH*; Hs02758991_g1).

## Statistics and reproducibility

Normal distribution of data was tested by Kolmogorov-Smirnov and Lilliefors test as well as Shapiro-Wilk and D'Agostino & Pearson normality test. A non-parametric test was used when normality was rejected significantly, or sample size (n) was below 6. When comparing ≥ three groups, One-way ANOVA with Tukey multiple comparison test or Kruskal-Wallis ANOVA with Dunn multiple comparison test were performed. For comparison of two groups an unpaired students t-test or a Mann-Whitney test was used. A *p*-value of <0.05 was considered statistically

significant. Data are presented as mean ± SEM. Statistical analysis was performed using GraphPad Prism 10.1.2. A positive control (8-MOP) was used in all experiments and was run in parallel. No statistical method was used to predetermine the sample size. No data were excluded from the analyses. The experiments were not randomized. The investigators were not blinded when performing and analyzing histological immunostaining, western blot and quantitative PCR. Experiments were conducted in at least three independent biological replicate (human skin biopsies). Skin biopsies were retrieved from *n* = 9 independent body donors. Exact n-numbers are given within the figure legend. Western Blot and TaqMan PCRs were performed in two biopsy samples per body donor, using technical duplicates for TaqMan PCRs. The mean value of these two replicates represent the value for each body donor. Reproducibility was ensured by verifying consistent results across all replicates and using 8-MOP as positive control. Representative images were selected to display the best match of the graphically presented mean values of the data.

## Reporting summary

Further information on research design is available in the Nature Portfolio Reporting Summary linked to this article.

## Data availability

All data supporting the findings of this study are presented either in the paper or the Supplementary Information. Additionally, all numerical source data underlying the graphs presented are provided in Supplementary Data 1. Supplementary Fig. S13–S28 include uncropped and unedited Western blot images for the presented figures.

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

## Acknowledgements

We sincerely thank those who donated their bodies to science for anatomical research. Deutsche Forschungsgemeinschaft (DFG, German Research Foundation) Project-ID 322900939, (SFB TRR219 S-02). We gratefully acknowledge Nina Rebmann, Jeannette Zimolong and Ellen Becker for excellent technical support.

## Author contributions

M.H., S.J., F.G., F.M., M.B., J.M.F., M.T. conceived and designed research. M.H., S.J. conducted experiments and analyzed/interpreted data. L.W. and M.R.M. performed quantification of HCT in human skin biopsies using ultra-high-performance liquid chromatography and high-resolution tandem mass spectrometry (LC-HRMS/MS). TT organized acquisition of body donors. JR provided 8-MOP and UVP-crosslinker. M.H., F.G., F.M., M.B. and L.W. wrote the manuscript. M.B., F.M., T.T., P.B., M.T., M.R.M., J.M.F., J.R. critically interpreted the data, reviewed the manuscript.

## Funding

## Competing interests

M.H., S.J., L.W., P.B., T.T., M.T., M.R.M. declare no competing interests. These authors declare the following competing interests: F.G. has received speaker honoraria from AstraZeneca and has been supported by the German Heart Foundation (F/61/21). J.M.F. received funding of the German Cardiology Society during parts of the study (project number: 04/2022). F.M. is supported by Deutsche Gesellschaft für Kardiologie (DGK), Deutsche Forschungsgemeinschaft (SFB TRR219, Project-ID 322900939), and Deutsche Herzstiftung. Saarland University has received scientific support from Ablative Solutions, Medtronic and ReCor Medical. Until May 2024, F.M.

has received speaker honoraria/consulting fees from Ablative Solutions, Amgen, Astra-Zeneca, Bayer, Boehringer Ingelheim, Inari, Medtronic, Merck, ReCor Medical, Servier, and Terumo. MB reports personal fees from Amgen, Astra Zeneca, Bayer, Boehringer Ingelheim, Cytokinetics, Edwards, Medtronic, Novartis, Servier, and Vifor. JR received funding from the Jörg Wolff foundation.
