## [Transparent Peer Review file · Communications Biology]

Assessing Phototoxic Drug Properties of Hydrochlorothiazide Using Human Skin Biopsies

Corresponding Author: Dr Mathias Hohl

This manuscript has been previously reviewed at another journal. This document only contains information relating to versions considered at Communications Biology.

Version 0:

Reviewer comments:

Reviewer #2

(Remarks to the Author)

This study presents a novel human skin model to assess the photocarcinogenic effects of HCT as an alternative to the animal model. By using human skin biopsy specimens, the authors were able to differentiate the photodynamic effects of HCT on p53 activation and MDM2 gene expression in human skin exposed to UVA and UVB. Their findings suggest that UVA may activate the p53-MDM2 axis, potentially influencing skin carcinogenesis in HCT-treated skin.

While this study is intriguing, several important issues should be addressed. Below are some questions and comments:

Introduction

Both γ -H2A.X and γ H2A.X are mixedly used as abbreviations throughout the manuscript. Please correct this inconsistency.

Methods

Skin Specimens

The skin donors were quite elderly (approximately 87 years old), meaning their skin had likely been exposed to other carcinogenic stimuli (such as smoking or chemical exposure). Did the authors consider the medications these donors were using? Additionally, what skin type (Fitzpatrick classification) was used in this study? Please clarify the rationale behind selecting these specific donors and explain why the hip was chosen as the biopsy site.

UV Irradiation

It has been suggested that HCT-associated carcinogenesis requires prolonged UV exposure. However, the authors used a single exposure of 0.3 J/cm² UV irradiation in this study. Can a one-time UV exposure truly induce HCT-associated carcinogenesis? Please provide a rationale for selecting this irradiation protocol.

Results and Discussion

Although the study demonstrates that a single UVA exposure can activate the p53-MDM2 axis in HCT-treated skin, it remains unclear whether this activation promotes carcinogenesis, as no DNA damage was detected in the treated skin. Are there additional findings or evidence that suggest repeated activation of the p53-MDM2 axis could lead to carcinogenesis? This issue is critical and should be addressed in more depth.

In the Discussion, the authors mention that previous pharmacoepidemiologic studies are observational and non-randomized, raising concerns about potential biases. However, recent population-based cohort studies—some using propensity score-matched cohorts to simulate randomized controlled trials—are considered more credible than small RCTs. While variations in study outcomes are understandable, the association between HCT and carcinogenesis is widely accepted in most Western countries and has gained global recognition, including in Asia (Hashizume et al. JAAD int 2023). Therefore, the authors' concerns may be overstated.

Reviewer #3

(Remarks to the Author)

Assessing phototoxic drug properties of hydrochlorothiazide using human skin biopsies

In the manuscript entitled „Assessing phototoxic drug properties of hydrochlorothiazide using human skin biopsies,, the Authors presented an interesting alternative to conducting skin phototoxicity testing using human skin biopsies. Although some points in this manuscript require additional clarification:

1.The first step in the decision tree aimed at assessing the phototoxic potential of

a compound is the determination of its absorption spectrum in the UV-visible range. Such a spectrum would be welcome in this study, and is missing.

2.In this study, the Authors used UVA and UVB radiation at a dose of 300 mJ/cm². While it can be concluded from the literature that studies using radiation in the UVB range at the indicated dose may show an effect, I would ask you to justify the choice of UVA dose. Current FDA and EMA recommendations suggest conducting drug phototoxicity studies at doses of 5-20 J/cm². Therefore, I ask you to justify the choice of UVA radiation in the range of 0.3 J/cm², as in my opinion the chosen dose is insufficient to conduct a drug phototoxicity tests. It is likely that the obtained results are the effect of using too low a dose of UVA radiation.

3.Please justify why the Authors chose to conduct drug phototoxicity studies on UVA and UVB at the same doses when they do not reach the skin to the same extent under in vivo conditions? How do the chosen UVA and UVB doses relate to in vivo conditions?

4.As the Authors suggest that the presented model for drug phototoxicity testing better reflects in vivo conditions, please specify in the manuscript what methods of phototoxicity testing are available to date? The model including multiple drug administration and multiple radiation exposures seems to more closely reflect in vivo conditions.

5. Please clarify whether the use of a skin biopsy model taken post-mortem affects the efficiency of your research when normal skin function has ceased and it has no connection with the rest of the body.

6.Due to the fact that the phototoxic reaction is directly connected with the generation of reactive oxygen species, I suggest the determination of ROS in the analysed tissue in order to check whether a phototoxic reaction has occurred. The lack of induction of an inflammatory response following UVA exposure may be due to the use of too low a dose of UVA. Please include a description of the phototoxic reaction in the introduction.

7.Establish a stronger link between hypertension and the phototoxic effects of HCT. Discuss the relevance of these findings to individuals with hypertension and the potential impact on their overall health.

8.Expand on the discussion regarding drug-induced photosensitivity. Provide additional context on why this is a growing dermatological problem and how it impacts patient management and treatment decisions.

Version 1:

Reviewer comments:

Reviewer #2

(Remarks to the Author)

This manuscript has been thoroughly revised in response to my comments. The experiments using higher-dose UV irradiation provide more robust evidence supporting HCT's role as a photosensitizer with potential carcinogenic effects. However, as the authors acknowledged, additional experiments involving repeated UV irradiation may be necessary to establish this effect fully.

Reviewer #3

(Remarks to the Author)

The Authors responded to the comments in the review and performed additional experiments.

I believe that in its current version the manuscript should be published.

I thank the Authors for their very detailed responses.

We would like to thank the editors and reviewers for their thoughtful comments, which further aided us in improving the manuscript. Please find attached a point-by-point response to your comments. To address the reviewers valid comments, we conducted additional experiments and implemented the novel results within the revised manuscript. For visualization, all changes have been marked in red throughout the manuscript.

Reviewers' comments:

Reviewer #2 (Remarks to the Author):

This study presents a novel human skin model to assess the photocarcinogenic effects of HCT as an alternative to the animal model. By using human skin biopsy specimens, the authors were able to differentiate the photodynamic effects of HCT on p53 activation and MDM2 gene expression in human skin exposed to UVA and UVB. Their findings suggest that UVA may activate the p53-MDM2 axis, potentially influencing skin carcinogenesis in HCT-treated skin.

While this study is intriguing, several important issues should be addressed. Below are some questions and comments:

Introduction

Both γ -H2A.X and γ H2A.X are mixedly used as abbreviations throughout the manuscript. Please correct this inconsistency.

Thank you. The inconsistency has been corrected.

Methods

Skin Specimens

The skin donors were quite elderly (approximately 87 years old), meaning their skin had likely been exposed to other carcinogenic stimuli (such as smoking or chemical exposure). Did the authors consider the medications these donors were using? Additionally, what skin type (Fitzpatrick classification) was used in this study? Please clarify the rationale behind selecting these specific donors and explain why the hip was chosen as the biopsy site.

We concur with the reviewer that information about life-style related carcinogenic stimuli or medical pretreatments of the body donors would be very interesting. Unfortunately, such information are not accessible due to protection of data privacy and ethical reasons. We used skin from the upper hip, as it is a part of the body that likely was not exposed to sunlight regularly, reducing potential confounder based on UVA/B exposure during lifetime. The body donors were Caucasians with a skin-type Fitzpatrick II-III.

We added the following sentence within the revised manuscript (Methods, page 15, line 23 to page 16, line 2)

“Skin biopsies (Six female, three male, median age: 84.56 ± 3.2 years, Caucasians with a skin type II-III on the Fitzpatrick scale) were collected <24 h after death from the upper hip of the body donor from skin areas that were macroscopic without pathological findings and without

signs of putrefaction. Skin from the upper hip was chosen to reduce potential confounders based on UV-exposure during life-time”

In our study, we used all-comers body donors with written informed consent during lifetime. Collecting skin biopsies from other cohorts like forensic medicine with access to younger individuals does not allow for sampling since the corpses are confiscated. Using skin from elderly donors more closely reflects older patients with hypertension who receive antihypertensive therapy. In line, patient cohorts in clinical studies investigating the risk of nonmelanoma skin cancer in HCT users were > 60 years of age (1, 2)

- 1) Hashizume H, Nakatani E, Sasaki H, Miyachi Y. Hydrochlorothiazide increases risk of nonmelanoma skin cancer in an elderly Japanese cohort with hypertension: The Shizuoka study. *JAAD Int.* 2023 Apr 26;12:49-57. doi: 10.1016/j.jdin.2023.04.007. PMID: 37274382; PMCID: PMC10236168.
- 2) Pedersen SA, Gaist D, Schmidt SAJ, Hölmich LR, Friis S, Pottegård A. Hydrochlorothiazide use and risk of nonmelanoma skin cancer: A nationwide case-control study from Denmark. *J Am Acad Dermatol.* 2018 Apr;78(4):673-681.e9. doi: 10.1016/j.jaad.2017.11.042. Epub 2017 Dec 4. PMID: 29217346.

UV Irradiation

It has been suggested that HCT-associated carcinogenesis requires prolonged UV exposure. However, the authors used a single exposure of 0.3 J/cm² UV irradiation in this study. Can a one-time UV exposure truly induce HCT-associated carcinogenesis? Please provide a rationale for selecting this irradiation protocol.

As outlined by the reviewer, carcinogenesis requires prolonged and recurring UV exposure. However, exposure to single UV irradiation might be sufficient to initiate acute phototoxic reactions, if photosensitizer, like drugs or their metabolites, are accumulated in the skin. Upon UV-light of the appropriate wave length the photosensitizer absorbs the light energy to form an excited triplet, which either reacts with oxygen to form free radicals or it covalently binds to tissue molecules (e.g. 8-MOP bind to DNA), both mechanisms causing cell damage (1). In case of HCT, UVA irradiation causes photodehalogenation and formation of a reactive form, which can damage DNA (2, 3). It was our aim to evaluate the potential phototoxicity of HCT in a new model of human body donor skin. In previous clinical trials we assessed changes in minimal erythema dose to UVA, which also occurs after a single irradiation (4). In this pilot study, the dose of 300 mJ/cm² was chosen as it reflects the lowest dose recommended in the German Guidelines for Phototherapy to induce phototoxic reactions in combination with psoralen (8-MOP), which was our positive control (5). Indeed, using 300 mJ/cm² UVA in 8-MOP-treated biopsies resulted in pronounced DNA-damage, elevated p53 protein levels, increased p53 phosphorylation and nuclear translocation of p53. Using 300 mJ/cm² UVA to irradiate HCT-treated biopsies resulted in comparable effects on p53 activation and stabilization as observed in the positive control, however in the absence of detectable DNA-damage, which was unexpected but intriguing novel finding that prompted us to continue with this irradiation protocol. However, as will be further addressed below, we adapted our irradiation protocol as suggested by reviewer 3 and as recommended by the United States Food and Drug Administration (6) and irradiated three additional skin biopsies with high-dose 5 J/cm² UVA as described in our protocol. We ask the reviewer to please see our novel results below.

- 1) Glatz M, Hofbauer GF. Phototoxic and photoallergic cutaneous drug reactions. *Chem Immunol Allergy.* 2012;97:167-79. doi: 10.1159/000335630. Epub 2012 May 3. PMID: 22613861.
- 2) Tamat SR, Moore DE. Photolytic decomposition of hydrochlorothiazide. *J Pharm Sci.* 1983 Feb;72(2):180-3. doi: 10.1002/jps.2600720221. PMID: 6834257.

- 3) Kreuz R, Algharably EAH, Douros A. Reviewing the effects of thiazide and thiazide-like diuretics as photosensitizing drugs on the risk of skin cancer. *J Hypertens* 2019;37:1950–1958.
- 4) Götzinger F, Hohl M, Lauder L, Millenaar D, Kunz M, Meyer MR, Ukena C, Lerche CM, Philipsen PA, Reichrath J, Böhm M, Mahfoud F. A randomized, placebo-controlled, trial to assess the photosensitizing, phototoxic and carcinogenic potential of hydrochlorothiazide in healthy volunteers. *J Hypertens* 2023;41:1853–1862
- 5) Herzinger T, Berneburg M, Ghoreschi K, Gollnick H, Hölzle E, Hönigsmann H, Lehmann P, Peters T, Röcken M, Scharffetter-Kochanek K, Schwarz T, Simon J, Tanew A, Weichenthal M. S1-Guidelines on UV phototherapy and photochemotherapy. *J Dtsch Dermatol Ges.* 2016 Aug;14(8):853-76. doi: 10.1111/ddg.12912. PMID: 27509435.
- 6) S10 Photosafety Evaluation of Pharmaceuticals Guidance for Industry. Available at: <https://www.fda.gov/downloads/drugs/guidances/ucm337572.pdf>. (Accessed 29 December 2018)

Results and Discussion

Although the study demonstrates that a single UVA exposure can activate the p53-MDM2 axis in HCT-treated skin, it remains unclear whether this activation promotes carcinogenesis, as no DNA damage was detected in the treated skin. Are there additional findings or evidence that suggest repeated activation of the p53-MDM2 axis could lead to carcinogenesis? This issue is critical and should be addressed in more depth.

We thank the reviewer for pointing out this critical issue. Besides a plethora of literature involving dysfunction of the p53-MDM2 axis with tumorigenesis the evidence that activation of the p53-MDM2 axis in the absence of DNA-damage by a single dose of UVA is sufficient to promote carcinogenesis is lacking. DNA-damage has been established as the major cause of skin cancer. As mentioned above, we collected new biopsies from the hip of 3 additional body donors (two female, one male) and repeated the UVA irradiation experiments according to our protocol but now using a higher dosage of 5 J/cm² in control, HCT- and 8-MOP treated biopsies.

*We observed, that six hours after irradiation with high dose 5 J/cm² UVA, p53 protein levels, and p53-phosphorylation was upregulated in HCT-treated skin biopsies. This was accompanied by a pronounced upregulation of UVA-induced DNA-damage, determined by increased γ H2A.X protein level. Histological immunofluorescence staining for p53 and γ H2A.X demonstrated an increased number of p53- and γ H2A.X positive nuclei in HCT-treated biopsies after UVA irradiation. Following 5 J/cm² UVA, phosphorylation of p38 MAPK was upregulated in HCT-treated biopsies as well as an increased mRNA expression of TNF α and CTGF (**Figure 5 revised manuscript; Supplementary Figure S8 revised manuscript**).*

Figure 5: UVA irradiation with 5 J/cm² induced p53 phosphorylation, nuclear translocation, γ H2A.X formation, and inflammatory response in HCT-treated skin biopsies after six hours.

(A) Representative Western blots demonstrating protein level of tumor suppressor protein p53, phosphorylation of histone H2A.X (γ H2A.X, Serin139) and phosphorylation of p53 (Serin15) in untreated control biopsies (Ctrl), and in biopsies treated with HCT six hours after irradiation with 5 J/cm² UVA. Unirradiated biopsies served as group-specific control (non). Protein expression of Glyceraldehyde 3-Phosphate Dehydrogenase (GAPDH) served as loading control. Quantification of (B) p53 protein, (C) phospho-p53 (Ser15), and (D) DNA-damage marker γ H2A.X. (E) Gene expression of p53-regulator MDM2 normalized against GAPDH six hours after irradiation (F) Representative double-stained immunohistochemistry for γ H2A.X and p53 of human skin biopsies six hours after irradiation with 5 J/cm² UVA. (G) Quantification of p53 positive stained nuclei and (H) γ H2A.X positive stained nuclei six hours after UVA irradiation in the epidermis of Ctrl or HCT-treated skin biopsies. (I) Representative Western blots demonstrating phosphorylation of p38 MAPK (T180/Y182) and total p38 MAPK protein in Ctrl, or HCT six hours after irradiation with 5 J/cm² UVA. Unirradiated biopsies served as group-specific control (non). (J) Quantification of phospho-p38 MAPK in relation to total p38 MAPK protein. (K) Gene expression of pro-inflammatory marker Tumor Necrosis Factor alpha (TNF α), Interleukin 6 (IL6), Interleukin 1 β , and (L) Connective Tissue Growth Factor (CTGF) γ normalized against GAPDH six hours after irradiation. For (B, C, D, E, G, H, J, K, L) n=3 per group. Data are shown as mean \pm SEM with individual points. For comparison of groups, P-value was determined using Kruskal-Wallis with Dunn's multiple comparisons test for all groups in (B, C, D, E, G, H, J, K, L). A Mann-Whitney test (#) was used for comparison of 2 groups for (D, H, J). IOD: Integrated optical density. Source data are provided as a Source Data file.

Online Supplementary Figure S8: Nuclear location of p53 and the DNA-damage marker γ H2A.X six hours after irradiation with 5J/cm² in Control (Ctrl) and HCT-treated skin biopsies. Representative immunofluorescence images of DAPI (nuclei in blue), p53 (green), γ H2A.X (red) and merged images of stainings (scale bar 50µm). White box shows scale bar 10µm:

Twenty-four hours after UVA irradiation, p53 protein levels as well as the amount of phosphorylated-p53 were upregulated in Ctrl and HCT-treated skin biopsies, while upregulation of DNA-damage marker protein γ H2A.X was restricted to HCT. Following UVA, activation of MDM2 mRNA expression of was more pronounced in HCT than in Ctrl.. Histological immunofluorescence stainings for p53 and γ H2A.X demonstrated an increased number of p53 positive nuclei in Ctrl and HCT-treated biopsies after UVA, however nuclei positive for γ H2A.X were detected only in HCT+UVA. In HCT, UVA-induced activation of p38 MAPK was accompanied with an increased mRNA expression of pro-inflammatory $TNF\alpha$, IL6, IL1 β as well as CTGF (Figure 6 revised manuscript; Supplementary Figure S9 revised manuscript).

Figure 6: UVA irradiation with 5 J/cm² induced p53 activation, pronounced γH2A.X formation and inflammatory response in HCT-treated skin biopsies after 24 hours

(A) Representative Western blots demonstrating protein level of tumor suppressor protein p53, phosphorylation of histone H2A.X (γH2A.X, Serin139) and phosphorylation of p53 (Serin15) in untreated control biopsies (Ctrl), and in biopsies treated with HCT 24 hours after irradiation with 5 J/cm² UVA. Unirradiated biopsies served as group-specific control (non). Protein expression of Glyceraldehyde 3-Phosphate Dehydrogenase (GAPDH) served as loading control. Quantification of (B) p53 protein, (C) phospho-p53 (Ser15), and (D) DNA-damage marker γH2A.X. (E) Gene expression of p53-regulator MDM2 normalized against GAPDH 24 hours after irradiation (F) Representative double-stained immunohistochemistry for γH2A.X and p53 of human skin biopsies 24 hours after 5 J/cm² UVA. (G) Quantification of p53 positive stained nuclei and (H) γH2A.X positive stained nuclei 24 hours after UVA irradiation in the epidermis of Ctrl or HCT-treated skin biopsies. (I) Representative Western blots demonstrating phosphorylation of p38 MAPK (T180/Y182) and total p38 MAPK protein in Ctrl, or HCT 24 hours after irradiation with 5 J/cm² UVA. Unirradiated biopsies served as group-specific control (non). (J) Quantification of phospho-p38 MAPK in relation to total p38 MAPK protein. (K) Gene expression of pro-inflammatory marker Tumor Necrosis Factor alpha (TNFα), Interleukin 6 (IL6), Interleukin 1β, and (L) Connective Tissue Growth Factor (CTGF) normalized against GAPDH 24 hours after irradiation. Data are shown as mean±SEM with individual points. For (B, C, D, E, G, H, J, K, L) $n=3$ per group. For comparison of groups P-value was determined using Kruskal-Wallis with Dunn's multiple comparisons test for (B, C, D, E, G, H, J, K, L). A Mann-Whitney test (#) was used for comparison of 2 groups for (D, H, J). IOD: Integrated optical density. Source data are provided as a Source Data file.

Online Supplementary Figure S9: Nuclear location of p53 and the DNA damage marker γ H2A.X 24 hours after irradiation with 5J/cm² in Control (Ctrl) and HCT-treated skin biopsies. Representative immunofluorescence images of DAPI (nuclei in blue), p53 (green), γ H2A.X (red) and merged images of stainings (scale bar 50 μ m). White box shows scale bar 10 μ m

We added these novel findings within the revised manuscript. *Results, page 7, line 16 to page 8 line 15:*

“Irradiation of HCT-treated skin biopsies with high dose UVA (5 J/cm²) results in stabilization of tumor suppressor p53, pronounced DNA-damage and activation of inflammatory response.

Six hours after irradiation with 5 J/cm² UVA, p53 protein stabilization and phosphorylation was induced, being more distinct in HCT-treated biopsies than in Ctrl. Following high dose UVA, expression of DNA-damage marker protein γ H2A.X was markedly increased in HCT-treated biopsies but not in Ctrl (1.25 \pm 0.18 vs. 9.79 \pm 2.4 IOD/GAPDH; $P=0.1000$ vs. Ctrl+UVA) (**Figure 5A-5D**). Gene expression of p53-regulator MDM2 was enhanced in Ctrl but not in HCT (**Figure 5E**). Histological immunofluorescence staining for p53 and γ H2A.X demonstrated an elevated number of p53-positive nuclei in Ctrl and HCT-treated biopsies after 5 J/cm² UVA irradiation, while γ H2A.X-positive nuclei were increased only in HCT but not in Ctrl (0.90 \pm 0.7 vs. 94.53 \pm 2.5 %; $P=0.1000$ vs. Ctrl+UVA) (**Figure 5E-5H, Online Supplements Figure S8**). Following 5 J/cm² UVA, increased phosphorylation of p38 MAPK

was accompanied by expression of TNF α and CTGF in HCT-treated biopsies but not in Ctrl (**Figure 5I-5L**).

Twenty-four hours after high-dose UVA irradiation, p53 protein, and phosphorylation of p53 were upregulated in Ctrl and, more pronounced, in HCT-treated skin biopsies. UVA-induced DNA damage, determined by elevated γ H2A.X protein expression, was restricted to HCT (1.38 \pm 0.25 vs. 6.91 \pm 1.9 IOD/GAPDH, $P=0.1000$ vs. Ctrl+UVA) (**Figure 6A-D**). Following UVA, expression of MDM2 mRNA was activated in HCT and in Ctrl. Histological immunofluorescence stainings for p53 and γ H2A.X demonstrated an increased number of p53-positive nuclei in Ctrl and HCT-treated biopsies after UVA, while nuclei positive for γ H2A.X were detected only in HCT+UVA (1.23 \pm 0.6 vs. 88.13 \pm 5.9 %, $P=0.1000$ vs. Ctrl+UVA) (**Figure 6F-6H, Online Supplements Figure S9**). In HCT but not in Ctrl, high dose UVA induced activation of p38 MAPK accompanied by an increased mRNA expression of pro-inflammatory TNF α , IL6, IL1 β , and the signaling molecule CTGF (**Figure 6I-6L**)."

*As mentioned above, all experiments were conducted in parallel using skin biopsies treated with 8-MOP as a positive control. Following irradiation with 5 J/cm² UVA, p53 protein levels, and p53 phosphorylation were elevated. This was accompanied by an upregulation of the DNA-damage marker γ H2A.X, while MDM2 mRNA was repressed (**Online Supplementary Figure S4 revised manuscript**). Increased nuclear translocation of p53 and increased DNA-damage in 8-MOP + UVA was confirmed histologically (**Online Supplementary Figure S5 revised manuscript**). Additionally, 5 J/cm² UVA triggered activation of p38 MAPK, upregulation of TNF α , and CTGF mRNA expression, while gene expression of interleukin 1 β was repressed (**Online Supplementary Figure S6 revised manuscript**). Gene expression of tumor-suppressors vitamin D receptor, purinergic receptor P2X7 were downregulated 24 hours after UVA irradiation (**Online Supplementary Table S2**).*

We added the following sentences within the revised manuscript: Results, page 5, line 2 to line 9:

"Following irradiation with high dose UVA (5 J/cm²), elevated p53 protein levels, p53 phosphorylation, and nuclear translocation was accompanied by an upregulation of the DNA-damage marker γ H2A.X, while MDM2 mRNA was repressed. Additionally, high-dose UVA triggered activation of p38 MAPK, upregulation of TNF α , and the regulatory signaling

molecule Connective Tissue Growth Factor (CTGF) mRNA expression, while transcription of tumor-suppressors vitamin D receptor, purinergic receptor P2X7 were downregulated. These effects were more pronounced after 24 hours (**Online Supplementary Figure S4-S6, Table S2).**”

In the Discussion, the authors mention that previous pharmacoepidemiologic studies are observational and non-randomized, raising concerns about potential biases. However, recent population-based cohort studies—some using propensity score-matched cohorts to simulate randomized controlled trials—are considered more credible than small RCTs. While variations in study outcomes are understandable, the association between HCT and carcinogenesis is widely accepted in most Western countries and has gained global recognition, including in Asia (Hashizume et al. JAAD int 2023). Therefore, the authors' concerns may be overstated.

Thank you for your comment. As described in our introduction, the possible associations of HCT and non-melanoma skin cancer caused changes in blood pressure medication prescriptions (i.e. stopping HCT without replacement) which presumably led to worsening blood pressure care in close to 80.000 patients in Germany alone. As hypertension is the most common cause of premature death and morbidity worldwide and HCT one of the most commonly used diuretics (especially in fixed-dose combinations), understanding possible adverse events is important. The pharmacoepidemiologic studies, although having some strengths (large number of patients, propensity score matching, dose-response-effects) remain biased (bias by indication, time window bias). Therefore, understanding the exact mechanisms of interaction between HCT and irradiation is of utmost importance. Randomized controlled trials and mechanistic studies in human tissue appear to be suitable for inducing such mechanisms. Additionally, from a clinical perspective, understanding the risk-benefit calculation of hypertension treatment vs. possibly developing a non-malignant skin tumor appears to be crucial.

The discussion has now been modified at page 13, line 12 to line 25:

“Drug-induced photosensitivity and phototoxicity is estimated to occur in 8% of all adverse drug reactions⁴. Since, sun tanning behavior and polypharmacy are increasingly practiced, these numbers might increase in the future. Patient management however can easily be impacted by advising sun protective behavior, like wearing of protective clothes, applying sunscreen and avoiding exposure when irradiation is the highest⁵¹. From a clinical perspective, the biggest impact of possible phototoxicity might be non-adherence to drugs because of fear of non-melanoma skin cancers². As stated above, hypertension remains the most common cause of premature death and morbidity worldwide and blood pressure control reduces the incidence of

stroke, heart failure, myocardial infarction and chronic kidney disease⁵². Not treating hypertension in fear of semi-malignant or non-malignant skin tumors might cause more harm than good to individuals. Moreover, phototoxicity might be easily avoidable by using protective clothes and sunscreen⁵². More evidence is needed to better understand the benefit from risk assessment between HCT, hypertension and non-melanoma skin cancers.“

Reviewer #3 (Remarks to the Author):

Assessing phototoxic drug properties of hydrochlorothiazide using human skin biopsies

In the manuscript entitled „Assessing phototoxic drug properties of hydrochlorothiazide using human skin biopsies,, the Authors presented an interesting alternative to conducting skin phototoxicity testing using human skin biopsies.

Although some points in this manuscript require additional clarification:

1. The first step in the decision tree aimed at assessing the phototoxic potential of a compound is the determination of its absorption spectrum in the UV-visible range. Such a spectrum would be welcome in this study, and is missing.

We totally agree that the determination of the absorption spectrum in the visible range would be welcome, however this approach would go far beyond the scope of this pilot study. Data concerning the UV-absorption spectrum of HCT was published previously by Kunisada et al. (1) and Selvaag et al. (2), demonstrating that HCT absorbs in the UVB-UVA range.

- (1) Kunisada M, Masaki T, Ono R, Morinaga H, Nakano E, Yogiarti F, Okunishi K, Sugiyama H, Nishigori C. Hydrochlorothiazide enhances UVA-induced DNA damage. Photochem Photobiol. 2013 May-Jun;89(3):649-54. doi: 10.1111/php.12048. Epub 2013 Feb 25. PMID: 23331297.*
- (2) Selvaag E, Petersen AB, Gniadecki R, Thorn T, Wulf HC. Phototoxicity to diuretics and antidiabetics in the cultured keratinocyte cell line HaCaT: evaluation by clonogenic assay and single cell gel electrophoresis Comet assay). Photodermatol Photoimmunol Photomed. 2002 Apr;18(2):90-5. doi: 10.1034/j.1600-0781.2002.180206.x. PMID: 12147042.*

2. In this study, the Authors used UVA and UVB radiation at a dose of 300 mJ/cm². While it can be concluded from the literature that studies using radiation in the UVB range at the indicated dose may show an effect, I would ask you to justify the choice of UVA dose. Current FDA and EMA recommendations suggest conducting drug phototoxicity studies at doses of 5-20 J/cm². Therefore, I ask you to justify the choice of UVA radiation in the range of 0.3 J/cm², as in my opinion the chosen dose is insufficient to conduct a drug phototoxicity tests. It is likely that the obtained results are the effect of using too low a dose of UVA radiation.

In previous clinical trials we assessed changes in minimal erythema dose to UVA, which also occurs after a single irradiation. The dose of 0.3 J/cm² was chosen as it reflects the lowest dose recommended by the German Guidelines for Phototherapy to induce phototoxic reactions in combination with psoralen, which was our positive control (1). The initial results presented herein are first-of-its kind experiments, attempting at establishing a novel method to assess phototoxicity of cardiovascular drugs. We aimed at firstly showing that the skin remains viable

under experimental conditions, that it shows reactions to UV irradiation, and that it is susceptible to combination therapy of UV irradiation and HCT therapy.

We thank the reviewer for the valid suggestion that the dosage of 300 mJ/cm² UVA might be too low to test the phototoxicity of HCT. Phototoxicity involves the activation of a photoinstable compound upon exposure to light. Subsequent ROS formation inflicts cellular and DNA damage (2). It has been reported, that oxidative stress and inflammatory responses are important factors contributing to potential skin cancer risk attributed to photosensitizing drugs (2). In our study, irradiation of HCT-treated biopsies with 300 mJ/cm² UVA was sufficient to activate the p53-MDM2 axis but DNA-damage and activation of p38 MAPK-induced inflammatory response was not detectable. Therefore, we collected new biopsies from the hip of 3 additional body donors (two female, one male) and repeated the UVA irradiation experiments according to the reviewer's suggestion and the current FDA recommendations (3) following our protocol using a higher dose of 5 J/cm² UVA in control, HCT- and 8-MOP treated biopsies.

- (1) Herzinger T, Berneburg M, Ghoreschi K, Gollnick H, Hölzle E, Hönigsman H, Lehmann P, Peters T, Röcken M, Scharffetter-Kochanek K, Schwarz T, Simon J, Tanew A, Weichenthal M. S1-Guidelines on UV phototherapy and photochemotherapy. *J Dtsch Dermatol Ges.* 2016 Aug;14(8):853-76. doi: 10.1111/ddg.12912. PMID: 27509435.
- (2) Kreuz R, Algharably EAH, Douros A. Reviewing the effects of thiazide and thiazide-like diuretics as photosensitizing drugs on the risk of skin cancer. *J Hypertens* 2019;37:1950–1958.
- (3) S10 Photosafety Evaluation of Pharmaceuticals Guidance for Industry. Available at: <https://www.fda.gov/downloads/drugs/guidances/ucm337572.pdf>. (Accessed 29 December 2018)

We observed, that six hours after irradiation with 5 J/cm² UVA, p53 protein levels, and p53-phosphorylation was upregulated in HCT-treated skin biopsies. This was accompanied by a pronounced DNA-damage, determined by increased γ H2A.X protein level. Histological immunofluorescence staining for p53 and γ H2A.X demonstrated a significantly increased number of p53- and γ H2A.X positive nuclei in HCT-treated biopsies after UVA irradiation. UVA induced DNA-damage was restricted to HCT and was not detectable in Ctrl.

*Following 5 J/cm² UVA, phosphorylation of p38 MAPK was upregulated in HCT-treated biopsies as well as an increased mRNA expression of TNF α and CTGF (**Figure 5 revised manuscript; Supplementary Figure S8 revised manuscript**).*

Figure 5: UVA irradiation with high-dose 5 J/cm² induced p53 phosphorylation, nuclear translocation, γH2A.X formation, and inflammatory response in HCT-treated skin biopsies after six hours.

(A) Representative Western blots demonstrating protein level of tumor suppressor protein p53, phosphorylation of histone H2A.X (γH2A.X, Serin139) and phosphorylation of p53 (Serin15) in untreated control biopsies (Ctrl), and in biopsies treated with HCT six hours after irradiation with 5 J/cm² UVA. Unirradiated biopsies served as group-specific control (non). Protein expression of Glyceraldehyde 3-Phosphate Dehydrogenase (GAPDH) served as loading control. Quantification of (B) p53 protein, (C) phospho-p53 (Ser15), and (D) DNA-damage marker γH2A.X. (E) Gene expression of p53-regulator MDM2 normalized against GAPDH six hours after irradiation (F) Representative double-stained immunohistochemistry for γH2A.X and p53 of human skin biopsies six hours after irradiation with 5 J/cm² UVA. (G) Quantification of p53 positive stained nuclei and (H) γH2A.X positive stained nuclei six hours after UVA irradiation in the epidermis of Ctrl or HCT-treated skin biopsies. (I) Representative Western blots demonstrating phosphorylation of p38 MAPK (T180/Y182) and total p38 MAPK protein in Ctrl, or HCT six hours after irradiation with 5 J/cm² UVA. Unirradiated biopsies served as group-specific control (non). (J) Quantification of phospho-p38 MAPK in relation to total p38 MAPK protein. (K) Gene expression of pro-inflammatory marker Tumor Necrosis Factor alpha (TNFα), Interleukin 6 (IL6), Interleukin 1β, and (L) Connective Tissue Growth Factor (CTGF) normalized against GAPDH six hours after irradiation. For (B, C, D, E, G, H, J, K, L) n=3 per group. Data are shown as mean±SEM with individual points. For comparison of groups, P-value was determined using Kruskal-Wallis with Dunn's multiple comparisons test for all groups in (B, C, D, E, G, H, J, K, L). A Mann-Whitney test (#) was used for comparison of 2 groups for (D, H, J). IOD: Integrated optical density. Source data are provided as a Source Data file.

Online Supplementary Figure S8: Nuclear location of p53 and the DNA-damage marker γ H2A.X six hours after irradiation with 5J/cm² in Control (Ctrl) and HCT-treated skin biopsies. Representative immunofluorescence images of DAPI (nuclei in blue), p53 (green), γ H2A.X (red) and merged images of stainings (scale bar 50μm). White box shows scale bar 10μm:

Twenty-four hours after UVA irradiation, p53 protein levels as well as the amount of phosphorylated-p53 were upregulated in Ctrl and HCT-treated skin biopsies, while upregulation of DNA-damage marker γ H2A.X was restricted to HCT. Following UVA, activation of MDM2 mRNA expression was more pronounced in HCT compared to Ctrl. Histological immunofluorescence stainings for p53 and γ H2A.X demonstrated an increased number of p53 positive nuclei in Ctrl and HCT-treated biopsies after UVA, however nuclei positive for γ H2A.X were detected only in HCT+UVA. In HCT but not in Ctrl, UVA-induced activation of p38 MAPK was accompanied with an increased mRNA expression of pro-inflammatory TNF α , IL6, IL1 β as well as CTGF (Figure 6 revised manuscript; Supplementary Figure S9 revised manuscript).

Figure 6: UVA irradiation with high-dose 5 J/cm² induced p53 activation, pronounced γ H2A.X formation and inflammatory response in HCT-treated skin biopsies after 24 hours

(A) Representative Western blots demonstrating protein level of tumor suppressor protein p53, phosphorylation of histone H2A.X (γ H2A.X, Serin139) and phosphorylation of p53 (Serin15) in untreated control biopsies (Ctrl), and in biopsies treated with HCT 24 hours after irradiation with 5 J/cm² UVA. Unirradiated biopsies served as group-specific control (non). Protein expression of Glyceraldehyde 3-Phosphate Dehydrogenase (GAPDH) served as loading control. Quantification of (B) p53 protein, (C) phospho-p53 (Ser15), and (D) DNA-damage marker γ H2A.X. (E) Gene expression of p53-regulator MDM2 normalized against GAPDH 24 hours after irradiation (F) Representative double-stained immunohistochemistry for γ H2A.X and p53 of human skin biopsies 24 hours after 5 J/cm² UVA. (G) Quantification of p53 positive stained nuclei and (H) γ H2A.X positive stained nuclei 24 hours after UVA irradiation in the epidermis of Ctrl or HCT-treated skin biopsies. (I) Representative Western blots demonstrating phosphorylation of p38 MAPK (T180/Y182) and total p38 MAPK protein in Ctrl, or HCT 24 hours after irradiation with 5 J/cm² UVA. Unirradiated biopsies served as group-specific control (non). (J) Quantification of phospho-p38 MAPK in relation to total p38 MAPK protein. (K) Gene expression of pro-inflammatory marker Tumor Necrosis Factor alpha (TNF α), Interleukin 6 (IL6), Interleukin 1 β , and (L) Connective Tissue Growth Factor (CTGF) normalized against GAPDH 24 hours after irradiation. Data are shown as mean \pm SEM with individual points. For (B, C, D, E, G, H, J, K, L) n=3 per group. For comparison of groups P-value was determined using Kruskal-Wallis with Dunn's multiple comparisons test for (B, C, D, E, G, H, J, K, L). A Mann-Whitney test (#) was used for comparison of 2 groups for (D, H, J). IOD: Integrated optical density. Source data are provided as a Source Data file.

Online Supplementary Figure S9: Nuclear location of p53 and the DNA damage marker γ H2A.X 24 hours after irradiation with 5J/cm² in Control (Ctrl) and HCT-treated skin biopsies. Representative immunofluorescence images of DAPI (nuclei in blue), p53 (green), γ H2A.X (red) and merged images of stainings (scale bar 50 μ m). White box shows scale bar 10 μ m

We added these novel findings within the revised manuscript. *Results, page 7, line 16 to page 8 line 4:*

“Irradiation of HCT-treated skin biopsies with high dose UVA (5 J/cm²) results in stabilization of tumor suppressor p53, pronounced DNA-damage and activation of inflammatory response.

Six hours after irradiation with 5 J/cm² UVA, p53 protein stabilization and phosphorylation was induced, being more distinct in HCT-treated biopsies than in Ctrl. Following high dose UVA, expression of DNA-damage marker protein γ H2A.X was markedly increased in HCT-treated biopsies but not in Ctrl (1.25 \pm 0.18 vs. 9.79 \pm 2.4 IOD/GAPDH; $P=0.1000$ vs. Ctrl+UVA) (**Figure 5A-5D**). Gene expression of p53-regulator MDM2 was enhanced in Ctrl but not in HCT (**Figure 5E**). Histological immunofluorescence staining for p53 and γ H2A.X demonstrated an elevated number of p53-positive nuclei in Ctrl and HCT-treated biopsies after 5 J/cm² UVA irradiation, while γ H2A.X-positive nuclei were increased only in HCT but not in Ctrl (0.90 \pm 0.7 vs. 94.53 \pm 2.5 %; $P=0.1000$ vs. Ctrl+UVA) (**Figure 5E-5H, Online**

Supplementary Figure S8). Following 5 J/cm² UVA, increased phosphorylation of p38 MAPK was accompanied by expression of TNF α and CTGF in HCT-treated biopsies but not in Ctrl (**Figure 5I-5L**).

Twenty-four hours after high-dose UVA irradiation, p53 protein, and phosphorylation of p53 were upregulated in Ctrl and, more pronounced, in HCT-treated skin biopsies. UVA-induced DNA damage, determined by elevated γ H2A.X protein expression, was restricted to HCT (1.38 \pm 0.25 vs. 6.91 \pm 1.9 IOD/GAPDH, $P=0.1000$ vs. Ctrl+UVA) (**Figure 6A-D**). Following UVA, expression of MDM2 mRNA was activated in HCT and in Ctrl. Histological immunofluorescence stainings for p53 and γ H2A.X demonstrated an increased number of p53-positive nuclei in Ctrl and HCT-treated biopsies after UVA, while nuclei positive for γ H2A.X were detected only in HCT+UVA (1.23 \pm 0.6 vs. 88.13 \pm 5.9 %, $P=0.1000$ vs. Ctrl+UVA) (**Figure 6F-6H, Online Supplementary Figure S9**). In HCT but not in Ctrl, high dose UVA induced activation of p38 MAPK accompanied by an increased mRNA expression of pro-inflammatory TNF α , IL6, IL1 β , and the signaling molecule CTGF (**Figure 6I-6L**).”

Additionally, our novel data demonstrated that 5 J/cm² UVA resulted in a repressed gene expression of vitamin D receptor and P2X7 receptor after 6 hours in HCT-treated biopsies (Online Supplemental Table 3). Downregulation of these marker were connected with of tumorigenesis (1, 2)

- 1) Ellison TI, Smith MK, Gilliam AC, MacDonald PN. Inactivation of the vitamin D receptor enhances susceptibility of murine skin to UV-induced tumorigenesis. *J Invest Dermatol* 2008;128:2508–2517.
- 2) Geraghty NJ, Watson D, Adhikary SR, Shuyter R. P2X7 receptor in skin biology and diseases. *World J Dermatol* 2016;5:72.

We incorporated these results within the revised manuscript. Results, page 8, line 20 to page 9, line 10:

“Dose dependent effect of UVA on the expression of pro-apoptotic and carcinogenesis marker in HCT-treated skin biopsies

Low dose UVA (300mJ/cm²) had no effect on the expression of the vitamin D receptor or P2X7 receptor, neither in Ctrl nor in HCT (**Table 1**). However, in HCT-treated biopsies, irradiation

with high dose UVA (5 J/cm²) resulted in downregulation of vitamin D receptor mRNA (1.09±0.15 vs. 0.48±0.03 relative gene expression; *P*=0.5264 vs. Ctrl+UVA) and P2X7 receptor mRNA (1.60±0.21 vs. 0.38±0.07 relative gene expression; *P*=0.0389 vs. Ctrl+UVA) after 6 hours. The observed UVA-induced effects were weakened after 24 hours (**Online Supplementary Table S3**). Interestingly, pro-apoptotic Bax mRNA demonstrated a differential gene expression pattern depending on UVA-dosage. Six hours after low dose UVA-exposure, Bax mRNA was upregulated in HCT compared to Ctrl (1.10±0.1 vs. 1.87±0.3 relative gene expression/GAPDH, *P*=0.1508 vs. Ctrl+UVA; n=5/group), while Bax gene expression was significantly lowered in HCT after irradiation with 5 J/cm² compared to Ctrl (1.48±0.14 vs. 0.73±0.12 relative gene expression/GAPDH; *P*=0.0191 vs. Ctrl+UVA; n=3/group). Following UVB irradiation, vitamin D receptor and P2X7 receptor gene expression was significantly repressed in Ctrl and HCT, with no additive effect of HCT. Anti-apoptotic Bcl-2, pro-apoptotic Bak-1 and Bax mRNA were unaffected by UVB irradiated (**Table 1**).”

We added the following table as an Online Supplementary Table S3:

Online Supplementary Table S3: Effect of irradiation with high-dose 5 J/cm² on mRNA expression of marker genes involved in apoptosis and tumorigenesis

	Ctrl (n=3)	Ctrl +UVA (n=3)	HCT (n=3)	HCT +UVA (n=3)
Relative gene expression/GAPDH	6 hours after irradiation with 5 J/cm ²			
Vitamin D- Receptor	1.00±0.003	1.09±0.15	1.01±0.02	0.48±0.03
P2X7- Receptor	1.02±0.01	1.60±0.21	1.42±0.05	0.38±0.07 P =0.0389 vs. Ctrl+UVA
Bcl-2	1.01±0.01	1.13±0.13	1.09±0.10	0.84±0.10
Bak1	1.00±0.003	1.21±0.15	1.22±0.06	0.47±0.02
Bax	1.00±0.003	1.48±0.14	1.16±0.09	0.73±0.12 P =0.0191 vs. Ctrl+UVA

	24 hours after irradiation with 5 J/cm ²			
Vitamin D-Receptor	1.01±0.008	0.90±0.11	0.90±0.05	0.61±0.15
P2X7-Receptor	1.02±0.01	0.75±0.06	0.93±0.02	0.38±0.10
Bcl-2	1.01±0.005	0.67±0.03	0.92±0.12	0.70±0.08
Bak1	1.00±0.001	1.16±0.06	0.98±0.19	1.13±0.4
Bax	1.02±0.01	1.38±0.07	1.16±0.32	1.69±0.47

Data are shown as mean±SEM. For Vitamin D-Receptor, P2X7-Receptor, Bcl-2, Bak1, and Bax $n=3$ per group. Data are shown as mean±SEM. For comparison of groups P-value was determined using Kruskal-Wallis with Dunn's multiple comparisons test. Source data are provided as a Source Data file

*As mentioned within the manuscript, all experiments were conducted in parallel using skin biopsies treated with 8-MOP as a positive control. Following irradiation with 5 J/cm² UVA, p53 protein levels, and p53 phosphorylation were elevated. This was accompanied by an upregulation of the DNA-damage marker γ H2A.X, while MDM2 mRNA was repressed (**Online Supplementary Figure S4 revised manuscript**). Increased nuclear translocation of p53 and increased DNA-damage in 8-MOP + UVA was confirmed histologically (**Online Supplementary Figure S5 revised manuscript**). Additionally, 5 J/cm² UVA triggered activation of p38 MAPK, upregulation of TNF α , and CTGF mRNA expression, while gene expression of interleukin 1 β was repressed (**Online Supplementary Figure S6 revised manuscript**). Gene expression of tumor-suppressors vitamin D receptor, purinergic receptor P2X7 were downregulated 24 hours after UVA irradiation (**Online Supplementary Table S2**).*

Online Supplementary Figure S4: Effect of irradiation with 5 J/cm² UVA on p53 protein, p53 phosphorylation, γH2A.X, and expression of MDM2 mRNA in 8-MOP treated skin biopsies

(A) Representative Western blots demonstrating protein level of tumor suppressor protein p53, phosphorylation of histone H2A.X (γH2A.X) and phosphorylation of p53 (Serin15) in biopsies treated with 8-MOP (positive control) 6 hours after irradiation with 5 J/cm² UVA. Unirradiated biopsies served as group-specific control (non). Protein expression of Glyceraldehyde 3-Phosphate Dehydrogenase (GAPDH) served as loading control. Quantification of (B) p53 protein (C) phospho-p53, and (D) γH2A.X. (E) Gene expression of p53-regulator MDM2 normalized against GAPDH 6 hours after irradiation. (F) Representative Western blots of p53, γH2A.X and phosphor-p53 (Serin15) in biopsies treated with 8-MOP (positive control) 24 hours after irradiation with 5 J/cm² UVA. Unirradiated biopsies served as group-specific control (non). Protein expression of GAPDH served as loading control. Quantification of (G) p53 protein (H) phospho-p53, and (I) γH2A.X. (J) Gene expression of MDM2 normalized against GAPDH 24 hours after irradiation. For (B, C, D, E, G, H, I, J) $n=3$ per group. Data are shown as mean±SEM with individual points. P-value was determined using a Mann-Whitney test for comparison of groups for (B, C, D, E, G, H, I, J). Integrated optical density. Source data are provided as a Source Data file.

Online Supplementary Figure S5: Effect of irradiation with 5 J/cm² on nuclear expression of γ H2A.X and p53 in 8-MOP-treated biopsies

(A) Representative immunostaining for γ H2A.X of human skin biopsies six and 24 hours after irradiation with 5 J/cm² UVA. **(B)** Representative immunostaining for tumor suppressor protein p53 of human skin biopsies six and 24 hours after irradiation with 5 J/cm² UVA **(C)** Quantification of γ H2A.X positive stained nuclei 6 hours and 24 hours after UVA irradiation in

the epidermis of 8-MOP treated skin biopsies. **(D)** Quantification of p53 positive stained nuclei 6 hours and 24 hours after UVA irradiation in the epidermis of 8-MOP treated skin biopsies. For **(C, D)** $n=3$ per group. Data are shown as mean \pm SEM with individual data points. Mann-Whitney test was used for comparison of groups for **(C, D)**. Source data are provided as a Source Data file.

Online Supplementary Figure S6: Effect of irradiation with 5 J/cm² UVA on p38 MAPK activation and pro-inflammatory response in 8-MOP-treated skin biopsies.

(A) Representative Western blots demonstrating phosphorylation of p38 MAPK (T180/Y182) and total p38 MAPK protein in biopsies treated with 8-MOP (positive control) 6 hours after irradiation with 5 J/cm² UVA. Unirradiated biopsies served as group-specific control (non). **(B)** Quantification of phospho-p38 MAPK in relation to total p38 MAPK protein. **(C)** Gene expression of pro-inflammatory marker Tumor Necrosis Factor alpha (TNF α), Interleukin 6 (IL6), Interleukin 1 β (IL1 β) and **(D)** Connective Tissue Growth Factor (CTGF) normalized against Glyceraldehyde 3-Phosphate Dehydrogenase (GAPDH) 6 hours after irradiation. **(E)** Representative Western blots demonstrating phosphorylation of p38 MAPK (T180/Y182) and total p38 MAPK protein in 8-MOP 24 hours after irradiation with 5 J/cm² UVA. Unirradiated biopsies served as group-specific control. **(F)** Quantification of phospho-p38 MAPK in relation to total p38 MAPK protein. **(G)** Gene expression of pro-inflammatory marker Tumor Necrosis Factor alpha (TNF α), Interleukin 6 (IL6), Interleukin 1 β (IL1 β) and **(H)** Connective Tissue Growth Factor (CTGF) normalized against GAPDH 24 hours after irradiation. Data are shown as mean \pm SEM with individual data points. For **(B, C, D, F, G, H)** $n=3$ per group. For comparison of 2 groups a Mann-Whitney test was used for **(B, C, D, F, G, H)**. Source data are provided as a Source Data file.

Online Supplementary Table S2: Effect of irradiation with high-dose 5 J/cm² on mRNA expression of marker genes involved in apoptosis and tumorigenesis in 8-MOP-treated skin biopsies

	8-MOP	8-MOP+UVA 5J/cm ²
Relative gene expression/GAPDH	6 hours after irradiation with 5 J/cm ²	
Vitamin D-Receptor	1.27±0.01	0.78±0.07
P2X7-Receptor	0.95±0.12	0.64±0.12
Bcl-2	0.89±0.03	0.80±0.06
Bak1	0.83±0.13	0.54±0.04
Bax	0.75±0.12	0.78±0.06
Relative gene expression/GAPDH	24 hours after irradiation with 5 J/cm ²	
Vitamin D-Receptor	0.90±0.05	0.61±0.15
P2X7-Receptor	0.76±0.15	0.30±0.04
Bcl-2	0.61±0.08	0.38±0.02
Bak1	0.91±0.08	0.36±0.12
Bax	1.00±0.04	0.69±0.12

Data are shown as mean±SEM. For Vitamin D-Receptor, P2X7-Receptor, Bcl-2, Bak1, and Bax *n*=3 per group. Data are shown as mean±SEM. For comparison of 2 groups a Mann-Whitney test was used. Source data are provided as a Source Data file

We added the following sentences within the revised manuscript: Results, page 5, line 2 to line 9:

“Following irradiation with high dose UVA (5 J/cm²), elevated p53 protein levels, p53 phosphorylation, and nuclear translocation was accompanied by an upregulation of the DNA-damage marker γ H2A.X, while MDM2 mRNA was repressed. Additionally, high-dose UVA triggered activation of p38 MAPK, upregulation of TNF α , and the regulatory signaling molecule Connective Tissue Growth Factor (CTGF) mRNA expression, while transcription of tumor-suppressors vitamin D receptor, purinergic receptor P2X7 were downregulated. These effects were more pronounced after 24 hours (**Online Supplementary Figure S4-S6, Table S2**).”

3. Please justify why the Authors chose to conduct drug phototoxicity studies on UVA and UVB at the same doses when they do not reach the skin to the same extent under in vivo conditions? How do the chosen UVA and UVB doses relate to in vivo conditions?

300 mJ/cm² UVA was chosen as it reflects the lowest dose recommended in the German Guidelines for Phototherapy to induce phototoxic reactions in combination with psoralen (8-MOP). Also, 300 mJ/cm² UVB is recommended by the German Guidelines for Phototherapy for skin type II (which mostly reflected our subjects).

We used 8-MOP treated biopsies as a positive control and indeed, we observed severe UVA (and UVB)-induced DNA damage, p53 stabilization/phosphorylation and activation of inflammatory response using 300 mJ/cm² (Online Supplements Figure S1-S3). Based on these observations we continued to use this dosage to address the question whether HCT has comparable phototoxic effects compared to 8-MOP and whether UVA is as effective as UVB in activating the photosensitizing properties of HCT and 8-MOP and whether HCT adds on the known direct phototoxic effect of UVB.

- (1) Herzinger T, Berneburg M, Ghoreschi K, Gollnick H, Hölzle E, Hönigsmann H, Lehmann P, Peters T, Röcken M, Scharffetter-Kochanek K, Schwarz T, Simon J, Tanew A, Weichenthal M. S1-Guidelines on UV phototherapy and photochemotherapy. *J Dtsch Dermatol Ges.* 2016 Aug;14(8):853-76. doi: 10.1111/ddg.12912. PMID: 27509435.

We addressed this point within the “Limitation and strength of this study” (page 14, line 8 to 13):

“Lower doses of radiation were used, as recommended in the German S1 Guidelines on UV phototherapy and photochemotherapy for PUVA therapy and as we observed reactions to lower doses of UVA in previous clinical trials (1, 2)“

- 1.) Herzinger T, Berneburg M, Ghoreschi K, Gollnick H, Hölzle E, Hönigsmann H, Lehmann P, Peters T, Röcken M, Scharffetter-Kochanek K, Schwarz T, Simon J, Tanew A, Weichenthal M. S1-Guidelines on UV phototherapy and photochemotherapy. *J Dtsch Dermatol Ges.* 2016 Aug;14(8):853-76. doi: 10.1111/ddg.12912. PMID: 27509435.
- (2) Göttinger F, Hohl M, Lauder L, Millenaar D, Kunz M, Meyer MR, Ukena C, Lerche CM, Philipsen PA, Reichrath J, Böhm M, Mahfoud F. A randomized, placebo-controlled, trial to assess the photosensitizing, phototoxic and carcinogenic potential of hydrochlorothiazide in healthy volunteers. *J Hypertens* 2023;41:1853–1862.

4. As the Authors suggest that the presented model for drug phototoxicity testing better reflects in vivo conditions, please specify in the manuscript what methods of phototoxicity testing are available to date? The model including multiple drug administration and multiple radiation exposures seems to more closely reflect in vivo conditions.

Based on the “Council Regulation (EC) No 440/2008 of 30 May 2008 laying down test methods pursuant to Regulation (EC) No 1907/2006 of the European Parliament and of the Council on the Registration, Evaluation, Authorisation and Restriction of Chemicals (REACH) (1)” of the European Union and the recommendations of United States Food and Drug Administration (FDA) (2) three methods to assess phototoxicity are described:

- 1.) In Vitro 3T3 NRU-Phototoxicity Test (PT)

- 2.) *Reactive Oxygen Species Assay for chemical photoreactivity*
- 3.) *In Vitro Phototoxicity Reconstructed Human Epidermis Phototoxicity test method*

The FDA guidelines (1), additionally, report on using the SKH1 (Hr/hr) albino hairless mouse model.

We included the following sentences within the revised manuscript: Conclusion, page 15, line 14 to line 17:

“The human skin biopsy model might also be used in combination with already established in vitro assays, like the highly sensitive 3T3 NRU-phototoxicity test, the Reactive Oxygen Species (ROS)-assay for chemical photoreactivity, or the Reconstructed Human Epidermis Phototoxicity test (1, 2)”

- 1) *Official Journal of the European Union, L 142, 31 May 2008*
- 2) *S10 Photosafety Evaluation of Pharmaceuticals Guidance for Industry. Available at: <https://www.fda.gov/regulatory-information/search-fda-guidance-documents/s10-photosafety-evaluation-pharmaceuticals>*

*The reviewer has another well-considered point and indeed, using our model it is conceivable to apply multiple drugs, provided drug-uptake can be verified by HPLC-analysis to proof accumulation within the skin. Additionally, based on own preparatory work, human skin biopsies are still responsive to UV-light after two weeks in cell culture (see reviewer only **Figure 1**), allowing prolonged expose to drugs and/or repeated exposure to UV radiation for up to one week to ensure proper function and metabolism of skin biopsies (1, 2).*

- 1) *Wester RC, Christoffel J, Hartway T, Poblete N, Maibach HI, Forsell J. Human cadaver skin viability for in vitro percutaneous absorption: storage and detrimental effects of heat-separation and freezing. Pharm Res. 1998 Jan;15(1):82-4. doi: 10.1023/a:1011904921318. PMID: 9487551.*
- 2) *Neil JE, Brown MB, Williams AC. Human skin explant model for the investigation of topical therapeutics. Sci Rep 2020;10:21192..*

Reviewer only Figure 1: Responsiveness of human skin biopsy after 2 weeks in cell culture medium. A) Representative Western Blot demonstrating p53 protein, phospho-p53 and γ H2A.X after irradiation with 300 mJ/cm² UVB or UVA. Unirradiated biopsies served as group-specific control (non). Protein expression of Glyceraldehyde 3-Phosphate Dehydrogenase (GAPDH) served as loading control. (B) Quantification of mRNA expression of MDM2, TNF α , IL6 and IL1 β normalized against GAPDH following irradiation with 300 mJ/cm² UVB or UVA. (C) Representative double-stained immunohistochemistry for γ H2A.X and p53 of HCT-treated skin biopsies after 300 mJ/cm² UVA or UVB irradiation.

We adapted the chapter limitation and strength of this study accordingly adding the following sentences on page 14, line 17 to 19:

“Administration of multiple drugs and repeated exposure to UV-radiation would more closely reflect in vivo conditions and are conceivable using human skin biopsies, if viability, structural integrity, and functional metabolism are maintained^{3,4}.

5. Please clarify whether the use of a skin biopsy model taken post-mortem affects the efficiency of your research when normal skin function has ceased, and it has no connection with the rest of the body.

The reviewer addresses a limiting point of our model. Cadaveric skin or skin biopsies in general, are detached from the central nervous system, from blood and lymphatic vessels, with vital effects on temperature, blood flow, supply with nutrients and oxygen, the disposal of metabolic waste products, influx of inflammatory cells, wound healing processes or angiogenesis/lymphangiogenesis the latter being involved in tumorigenesis (1, 2).

Nevertheless, we demonstrated that the skin retains the ability to respond on UVA and UVB irradiation by activating UV-related pathways (phosphorylation of H2A.X, p53 or p38 MAPK)

and gene expression (MDM2, P2X7, VitA, TNFa) comparable to published data on mice or cell culture experiments. For ethical reasons, it is not conceivable to retrieve skin biopsies from volunteer donor as a “non-cadaveric” control (given the multitude (>30) of biopsies needed or drug intake). As published previously, human cadaver skin has been used as a skin graft and has been shown to remain viable with proper function and metabolism for at least 8-9 days (3, 4).

- 1) *Huggenberger R, Detmar M. The cutaneous vascular system in chronic skin inflammation. J Investig Dermatol Symp Proc. 2011 Dec;15(1):24-32. doi: 10.1038/jidsymp.2011.5. PMID: 22076324; PMCID: PMC3398151.*
- 2) *Houghton BL, Meendering JR, Wong BJ, Minson CT. Nitric oxide and noradrenaline contribute to the temperature threshold of the axon reflex response to gradual local heating in human skin. J Physiol. 2006 May 1;572(Pt 3):811-20. doi: 10.1113/jphysiol.2005.104067. PMID: 16497714; PMCID: PMC1780012.*
- 3) *Wester RC, Christoffel J, Hartway T, Poblete N, Maibach HI, Forsell J. Human cadaver skin viability for in vitro percutaneous absorption: storage and detrimental effects of heat-separation and freezing. Pharm Res. 1998 Jan;15(1):82-4. doi: 10.1023/a:1011904921318. PMID: 9487551.*
- 4) *Neil JE, Brown MB, Williams AC. Human skin explant model for the investigation of topical therapeutics. Sci Rep 2020;10:21192.*

We extended the limitation chapter accordingly on page 14, line 20 to line 24

“Cadaveric skin or skin biopsies in general, are detached from the central nervous system, from blood and lymphatic vessels, with vital effects on temperature regulation, blood flow, supply with nutrients and oxygen, the disposal of metabolic waste products, influx of inflammatory cells, wound healing processes or angiogenesis/lymphangiogenesis, the latter being involved in tumorigenesis (1, 2).”

6. Due to the fact that the phototoxic reaction is directly connected with the generation of reactive oxygen species, I suggest the determination of ROS in the analysed tissue in order to check whether a phototoxic reaction has occurred. The lack of induction of an inflammatory response following UVA exposure may be due to the use of too low a dose of UVA. Please include a description of the phototoxic reaction in the introduction.

The generation of reactive oxygen species (ROS) during phototoxic reactions is indeed a highly interesting point. UVB and UVA irradiation induce the formation of ROS in cutaneous tissue [1, 2]. Free radicals are highly reactive with a half-life of only seconds [3, 4]. The imbalance between the production of reactive oxygen species and the antioxidant defense systems results in oxidative stress, which causes damage to cellular proteins and DNA [4]. The skin's enzymatic antioxidant defense comprises copper-zinc superoxide dismutase (SOD 1), manganese SOD (SOD 2) and catalase. SOD converts superoxide anion into hydrogen peroxide (H₂O₂), which is degraded by catalase into water (H₂O) [3,4]. These enzymes, which maintain a redox balance within cells, have been shown to be modulated by UVA and UVB irradiation in murine skin [5]. In order to address the reviewers valid point we analyzed the expression of proteins involved in the detoxification of ROS.

1. Kitazawa M, Podda M, Thiele J, Traber MG, Iwasaki K, Sakamoto K, Packer L. Interactions between vitamin E homologues and ascorbate free radicals in murine skin homogenates irradiated with ultraviolet light. *Photochem Photobiol.* 1997 Feb;65(2):355-65. doi: 10.1111/j.1751-1097.1997.tb08571.x. PMID: 9066312.
2. Scharffetter-Kochanek K, Wlaschek M, Brenneisen P, Schauen M, Blanduschun R, Wenk J. UV-induced reactive oxygen species in photocarcinogenesis and photoaging. *Biol Chem.* 1997 Nov;378(11):1247-57. PMID: 9426184.
3. Rubio CP, Cerón JJ. Spectrophotometric assays for evaluation of Reactive Oxygen Species (ROS) in serum: general concepts and applications in dogs and humans. *BMC Vet Res.* 2021 Jun 26;17(1):226. doi: 10.1186/s12917-021-02924-8. PMID: 34174882; PMCID: PMC8235564.
4. Juan CA, Pérez de la Lastra JM, Plou FJ, Pérez-Lebeña E. The Chemistry of Reactive Oxygen Species (ROS) Revisited: Outlining Their Role in Biological Macromolecules (DNA, Lipids and Proteins) and Induced Pathologies. *Int J Mol Sci.* 2021 Apr 28;22(9):4642. doi: 10.3390/ijms22094642. PMID: 33924958; PMCID: PMC8125527.
5. Sander CS, Chang H, Salzmann S, Müller CS, Ekanayake-Mudiyanselage S, Elsner P, Thiele JJ. Photoaging is associated with protein oxidation in human skin in vivo. *J Invest Dermatol.* 2002 Apr;118(4):618-25. doi: 10.1046/j.1523-1747.2002.01708.x. PMID: 11918707.
6. Shindo Y, Witt E, Han D, Packer L. Dose-response effects of acute ultraviolet irradiation on antioxidants and molecular markers of oxidation in murine epidermis and dermis. *J Invest Dermatol.* 1994 Apr;102(4):470-5. doi: 10.1111/1523-1747.ep12373027. PMID: 8151122.

We reanalyzed our protein-lysates of control, HCT- and 8-MOP-treated biopsies irradiated with 300mJ/cm² UVA and UVB. The revised **Online Supplements Figure S4** demonstrates that irradiation with 300mJ/cm² UVA or UVB did not result in a regulation of SOD1, SOD2 or catalase protein neither in control, HCT nor 8-MOP at 6 hours and 24 hours. As suggested by the reviewer, 300 mJ/cm² might be too low a dose. As mentioned above, we collected new biopsies from the hip of 3 additional body donors and repeated the UVA irradiation experiments using high dose 5 J/cm² in control, HCT- and 8-MOP treated biopsies and analyzed protein expression of SOD1, SOD2 and Catalase 6 hours and 24 hours after irradiation. Quantification of Western blots demonstrated that 5 J/cm² UVA did not induce a differential regulation of these three anti-oxidative defense proteins as depicted in **Supplemental Figure S10** within the revised Online Supplements.

Shindo et al (6) reported that reduction of catalase and superoxide dismutase activity is dose-dependent and decreased in the epidermis following ultraviolet irradiation at doses above 5 J/cm² (up to 25 J/cm²). Possibly, 5J/cm² may have still been too low a dosage to induce deactivation of ROS-detoxifying protein.

We added the following sentences within the revised Results, Page 7, line 12-line 14:

“Protein expression of anti-oxidative enzymes catalase, superoxide dismutase 1 (SOD 1) or SOD 2 was unchanged in all groups at all time-points (**Online Supplementary Figure S7**).”

We added the following sentences within the revised Results, Page 8, line 15-line 18:

“Irradiation with high-dose UVA had no effect on protein expression of anti-oxidative enzymes catalase, superoxide dismutase 1 (SOD 1) or SOD 2 in all groups at all time-points (**Online Supplementary Figure S10**).”

As suggested by the reviewer, we included a description of the phototoxic reaction within the introduction on page 3, line 7 to line 12.

“A phototoxic reaction is initiated upon exposure to UV-light, if photosensitizer, like drugs or their metabolites, are accumulated in the skin. Upon UV-light of the appropriate wavelength the photosensitizer absorbs the light energy to form an excited triplet, which either reacts with oxygen to form reactive oxygen species, or it covalently binds to tissue molecules. Both mechanisms causing cellular damages to membranes, lipids and DNA (1).”

- 1) *Glatz M, Hofbauer GF. Phototoxic and photoallergic cutaneous drug reactions. Chem Immunol Allergy. 2012;97:167-79. doi: 10.1159/000335630. Epub 2012 May 3. PMID: 2261386*

A6 hours after irradiation with 300mJ/cm²**B**6 hours
Catalase protein**C**6 hours
SOD 1 protein**D**6 hours
SOD 2 protein**E**24 hours after irradiation with 300mJ/cm²**F**24 hours
Catalase protein**G**24 hours
SOD 1 protein**H**24 hours
SOD 2 protein
Online Supplementary Figure S7: Effect of UV-irradiation on expression of anti-oxidative capacity proteins catalase, superoxide dismutase 1 (SOD 1) and SOD 2 after six and 24 hours.

(A) Representative Western blots demonstrating protein expression of catalase, SOD 1 and SOD 2 in untreated (Ctrl), HCT-treated and 8-MOP-treated (positive control) biopsies 6 hours after irradiation with 300 mJ/cm² UVA or UVB. Unirradiated biopsies served as group-specific control. **(B)** Quantification of catalase, **(C)** SOD 1 and **(D)** SOD 2 normalized against Glyceraldehyde 3-Phosphate Dehydrogenase (GAPDH) 6 hours after irradiation. **(E)** Representative Western blots demonstrating protein expression of catalase, SOD 1 and SOD 2 in untreated (Ctrl), HCT-treated and 8-MOP-treated (positive control) biopsies 24 hours after irradiation with 300 mJ/cm² UVA or UVB. Unirradiated biopsies served as group-specific control. **(F)** Quantification of catalase, **(G)** SOD 1 and **(H)** SOD 2 normalized against Glyceraldehyde 3-Phosphate Dehydrogenase (GAPDH) 24 hours after irradiation. For **B, C, D, F, G, H** n=4 per group. Data are shown as mean±SEM with individual data points. For comparison of three groups, P-value was determined using Kruskal-Wallis with Dunn's multiple comparisons test for all groups in **(B, C, D, F, G, H)** IOD: Integrated optical density. Source data are provided as a Source Data file.

Online Supplementary Figure S10: Effect of irradiation with high-dose 5 J/cm² UVA on expression of anti-oxidative capacity proteins catalase, superoxide dismutase 1 (SOD 1) and SOD 2 after six and 24 hours in untreated (Ctrl), HCT-treated and 8-MOP treated biopsies. (A) Representative Western blots demonstrating protein expression of catalase, SOD 1 and SOD 2 in untreated (Ctrl), HCT-treated and 8-MOP-treated (positive control) biopsies 6 hours after

irradiation with 5 J/cm² UVA. Unirradiated biopsies served as group-specific control. **(B)** Quantification of catalase, **(C)** SOD 1 and **(D)** SOD 2 normalized against Glyceraldehyde 3-Phosphate Dehydrogenase (GAPDH) 6 hours after irradiation. **(E)** Representative Western blots demonstrating protein expression of catalase, SOD 1 and SOD 2 in untreated (Ctrl), HCT-treated and 8-MOP-treated (positive control) biopsies 24 hours after irradiation with 5 J/cm² UVA. Unirradiated biopsies served as group-specific control. **(F)** Quantification of catalase, **(G)** SOD 1 and **(H)** SOD 2 normalized against Glyceraldehyde 3-Phosphate Dehydrogenase (GAPDH) 24 hours after irradiation. For **B, C, D, F, G, H** n=3 per group. Data are shown as mean±SEM with individual data points. IOD: Integrated optical density. For comparison of groups, P-value was determined using Kruskal-Wallis with Dunn's multiple comparisons test for all groups in **(B, C, D, F, G, H)**. Source data are provided as a Source Data file.

7. Establish a stronger link between hypertension and the phototoxic effects of HCT. Discuss the relevance of these findings to individuals with hypertension and the potential impact on their overall health.

Hypertension remains the most common cause of premature death and morbidity worldwide and blood pressure control reduces the incidence of stroke, heart failure, myocardial infarction and chronic kidney disease. Not treating hypertension in fear of semi-malignant or non-malignant skin tumors might cause more harm than good to individuals. Hypertension itself might be associated with cancer, independent of HCT therapy, although these interrelations are disputed. Presumably, common metabolic risk factors, like obesity and diabetes mellitus, might increase individuals' risk for the development of cancer and hypertension, if individuals are susceptible. From a clinical perspective, the biggest impact of possible phototoxicity might be non-adherence to drugs because of fear of non-melanoma skin cancers. Moreover, phototoxicity might be easily avoidable by using protective clothes and sunscreen. Please see question below.

8. Expand on the discussion regarding drug-induced photosensitivity. Provide additional context on why this is a growing dermatological problem and how it impacts patient management and treatment decisions.

We expanded the discussion as suggested by the reviewer and added the following sentences within the discussion (Page 13, line 12- line 25)

„Drug-induced photosensitivity and phototoxicity is estimated to occur in 8% of all adverse drug reactions¹. Since, sun tanning behavior and polypharmacy are increasingly practiced, these numbers might increase in the future. Patient management however can easily be impacted by advising sun protective behavior, like wearing of protective clothes, applying sunscreen and avoiding exposure when irradiation is the highest². From a clinical perspective, the biggest impact of possible phototoxicity might be non-adherence to drugs because of fear of non-melanoma skin cancers³. As stated above, hypertension remains the most common cause of

premature death and morbidity worldwide and blood pressure control reduces the incidence of stroke, heart failure, myocardial infarction and chronic kidney disease⁴. Not treating hypertension in fear of semi-malignant or non-malignant skin tumors might cause more harm than good to individuals. Moreover, phototoxicity might be easily avoidable by using protective clothes and sunscreen⁴. More evidence is needed to better understand the benefit from risk assessment between HCT, hypertension and non-melanoma skin cancers.

(1) Götzinger F, Reichrath J, Millenaar D, Lauder L, Meyer MR, Böhm M, Mahfoud F. Photoinduced skin reactions of cardiovascular drugs—a systematic review. *Eur Heart J Cardiovasc Pharmacother*. 2022 Jun 8;8(4):420-430. doi: 10.1093/ehjcvp/pvac017. PMID: 35278085.

(2) Sander M, Sander M, Burbidge T, Beecker J. The efficacy and safety of sunscreen use for the prevention of skin cancer. *CMAJ* 2020;192:E1802–E1808.

(3) Mahfoud F, Kieble M, Enners S, Werning J, Laufs U, Millenaar D, Böhm M, Kintscher U, Schulz M. 'Dear Doctor' warning letter (Rote-Hand-Brief) on hydrochlorothiazide and its impact on antihypertensive prescription. *Dtsch Arztebl Int* 2020;117:687–688.

(4) Jensen GB. Phototoxic and carcinogenic effects of hydrochlorothiazide: experimental study contrasting the pharmacoepidemiological evidence showing increased risk of skin cancer. *J Hypertens* 2023;41:1699–1700.